# Retrievit: In-context Retrieval Capabilities of Transformers, State Space Models, and Hybrid Architectures

**Georgios Pantazopoulos**                                              *gmp2000@hw.ac.uk*
*University of Edinburgh*
**Malvina Nikandrou**                                                   *mn2002@hw.ac.uk*
*University of Edinburgh*
**Ioannis Konstas**                                                     *i.konstas@hw.ac.uk*
*Heriot-Watt University*
**Alessandro Suglia**                                                   *asuglia@ed.ac.uk*
*University of Edinburgh*

**Reviewed on OpenReview:** *https://openreview.net/forum?id=PxkhbVeRRj*

## Abstract

Transformers excel at in-context retrieval but suffer from quadratic complexity with sequence length, while State Space Models (SSMs) offer efficient linear-time processing but have limited retrieval capabilities. We investigate whether hybrid architectures combining Transformers and SSMs can achieve the best of both worlds on two synthetic in-context retrieval tasks. The first task, n-gram retrieval, requires the model to identify and reproduce an n-gram that succeeds the query within the input sequence. The second task, position retrieval, presents the model with a single query token and requires it to perform a two-hop associative lookup: first locating the corresponding element in the sequence, and then outputting its positional index. Under controlled experimental conditions, we assess data efficiency, length generalization, robustness to out of domain training examples, and learned representations across Transformers, SSMs, and hybrid architectures. We find that hybrid models outperform SSMs and match or exceed Transformers in terms of data efficiency and extrapolation for tasks that require precise information retrieval from the input context. However, Transformers maintain superiority in position retrieval tasks. Through representation analysis, we discover that SSM-based models develop locality-aware embeddings where tokens representing adjacent positions become neighbors in embedding space, forming interpretable structures. This property is absent in Transformers as causal attention is sufficient for acquiring positional associations, and the introduction of positional encoding amplifies this behavior, leading to a consequent improvement in data efficiency. SSMs on the other hand update their internal representations incrementally and without positional encodings, are required to learn these associations. Our findings reveal fundamental differences in how Transformers and SSMs, and hybrid models learn positional associations[1].

## 1 Introduction

Transformers (Vaswani et al., 2017) have become the de facto option for sequence modeling due to their exceptional capabilities across a diverse range of applications, including natural language processing (Devlin et al., 2019), computer vision (Dosovitskiy et al., 2021) and multimodal applications (Tsimpoukelli et al., 2021; Radford et al., 2021). Despite their widespread success, Transformers are inherently constrained by certain architectural limitations primarily stemming from the self-attention mechanism's quadratic scaling with sequence length, which leads to substantial memory requirements and challenges with inference speed when processing long sequences.

---

[1]Code available at `https://github.com/gpantaz/retrievit`

This has driven significant interest in creating architectures that: 1) achieve similar performance as Transformers, 2) are efficient to train on modern hardware, and 3) require constant memory during inference. State Space Models (SSMs) (Gu et al., 2022; Goel et al., 2022; Poli et al., 2023; Gu & Dao, 2023; Dao & Gu, 2024) provide a promising alternative to Transformers as they can operate in a recursive mode that scales linearly with sequence length, avoiding the quadratic attention computation that bottlenecks Transformer training. This architectural difference also eliminates the memory explosion that occurs in attention-based models during inference, where key-value cache size grows proportionally with sequence length, enabling SSMs to process sequences of arbitrary length with constant-time autoregressive generation. Recent innovations in SSM design, such as Mamba (Gu & Dao, 2023) and Mamba2 (Dao & Gu, 2024), have further closed the performance gap with Transformers on language modeling benchmarks through selective scan mechanisms and hardware-aware parallelization strategies, while maintaining these computational advantages.

However, prior research (Jelassi et al., 2024; Merrill et al., 2024; De et al., 2024; Wen et al., 2025b) shows that these models have limited in-context retrieval capabilities. The ability to copy parts of the input to the output is fundamental for language models as it enables instruction following by generating contextually grounded responses (Ouyang et al., 2022), learning from in-context demonstrations (Brown et al., 2020), and accurate retrieval-augmented generation (Lewis et al., 2020). In particular, these works show that SSMs excel in tasks that require a summary of the inputs which can be effectively maintained in the hidden state, while Transformers maintain the lead in tasks requiring accessing precise parts from the context. Consequently, prior work tries to combine the two model families into hybrid architectures (Lenz et al., 2025; Ren et al., 2025; Blakeman et al., 2025), though the benefits of these models for in-context retrieval remain unclear.

In this work, we extend the prior findings regarding the in-context retrieval capabilities by examining the behavior of hybrid architectures on synthetic retrieval-oriented tasks that are indicative of a model's sequence modeling capabilities. More specifically, we view in-context retrieval from two prisms: 1) the ability to retrieve arbitrary information from the context (Jelassi et al., 2024), and 2) the ability to perform a two-hop association by matching the query to its position in the sequence (Pantazopoulos et al., 2024) (see Figure 1). The former is a proxy for the in-context capabilities of a model (Olsson et al., 2022). The latter has been applied to examine primarily multimodal sequence modeling capabilities (e.g., vision & text) (Zitkovich et al., 2023; Cheng et al., 2024), but also potentially relevant for any task where the inputs/outputs correspond to separate embedding spaces.

We focus on three aspects for assessing the quality of each model under controlled conditions, 1) *data efficiency:* how many samples are required for a model to learn the underlying task, 2) *length generalization:* from productive behavior, where we explore if a model can extend its predictions beyond the length it has seen in the training data (Hupkes et al., 2020; Newman et al., 2020; Pantazopoulos et al., 2022; Lee et al., 2025), 3) *robustness to ambiguous instances*, where we explore the behavior of each model to examples containing multiple correct candidate responses, and 4) *representation quality*, where we establish connections between the learnt representations and the structure of the task.

Through controlled comparisons between Transformers, SSMs, and hybrid architectures, we find that hybrid models outperform pure SSM models and have the capacity to outperform Transformers in terms of data efficiency and extrapolation when tasked to retrieve dense information from the context. However, Transformers maintain the lead in two-hop association compared to SSMs and hybrid models. We attribute this to a *locality-aware property* in models composed of SSM blocks. More specifically, we observed early during the training that models with SSM blocks tend to memorize positional information of tokens at the beginning and at the end of the sequence, and gradually learn the token-to-position mapping for intermediate tokens. On the other hand, Transformers learn the two-hop association task independent of the position within the sequence. Consequently, we project the representations of tokens that depict positions into lower dimensions and demonstrate that models with SSM blocks learn a locality-aware mapping, i.e., tokens that depict adjacent positions are neighbors within the embedding space. We show that this is a unique property of models with SSM as Transformers do not converge into such representations.

## 2 Related Work

**Synthetic tasks as probes for sequence modeling capabilities**  Synthetic tasks have been widely adopted as diagnostic tools for understanding the capabilities of a neural network (Kitaev et al., 2020; Olsson et al., 2022; Wang & Eisner, 2016; Arora et al., 2024; Nichani et al., 2025). In this work, we focus on the particular task of associative recall; a task of retrieving a stored memory or piece of information when given a related cue or partial input. Prior works use associative recall for benchmarking recurrent neural networks (Ba et al., 2016; Graves et al., 2014), where given a key that was previously paired with a value, the model must output the correct associated value. Recent advances in large language models have prompted a growing body of work linking model capabilities to associative recall mechanisms (Olsson et al., 2022; Arora et al., 2024; Wen et al., 2025b). As such, many variants of associative recall such as induction heads (Olsson et al., 2022), selective copying (Gu & Dao, 2023), n-gram retrieval (Jelassi et al., 2024), and synthetic factual recall (Nichani et al., 2025) establish links between synthetic task performance and real-world language modeling qualities. Relative to full-scale pre-training experiments, such synthetic benchmarks enable rapid architectural iteration while yielding theoretical insights into the fundamental limitations and scaling behaviors of diverse model architectures.

**Comparing Transformers and SSMs**  Several prior works study the key differences between Transformers and SSMs like Mamba for in-context learning (Akyürek et al., 2024; Grazzi et al., 2024), providing theoretical and empirical evidence on the limitations of SSMs in terms of expressivity (Muca Cirone et al., 2024), fixed-size memory (Merrill et al., 2024), and in-context retrieval (Jelassi et al., 2024; Pantazopoulos et al., 2024; Wen et al., 2025b). Collectively, these works show that SSMs like Mamba may perform on-par with Transformers on retrieval-oriented tasks whenever the underlying task relies on a summary of the inputs that can be effectively maintained in the hidden state. Meanwhile, Transformers naturally develop specialized "n-gram heads" (Akyürek et al., 2024), i.e., higher-order variants of induction heads (Olsson et al., 2022) that compute input-conditional next-token distributions resulting in advantages over other architectures. We adopt many concepts from prior work to quantify the retrieval capacity of hybrid models.

**Integrating Transformers and SSMs**  Several works try to integrate Transformers with SSMs resulting in hybrid architectures: 1) Interleaved models incorporate Transformer blocks where the objective of self-attention is to correct the linearly updated hidden state of the SSM blocks (Team et al., 2024; Dao & Gu, 2024; Glorioso et al., 2024; Lenz et al., 2025; Ren et al., 2025), 2) A two-stream approach (Dong et al., 2025), where the input at each block is processed individually by a Transformer and an SSM and fused together via a learnable gating mechanism. However, existing research predominantly focuses on ablation studies concerning the ratio of full SSM to attention layers, (Poli et al., 2023; Team et al., 2024; Lenz et al., 2025; Blakeman et al., 2025; Dong et al., 2025), frequently guided by tracking loss values, which is suitable for text modeling tasks but potentially overlooks the recall capabilities. In this work, we examine from a more critical prism the retrieval capabilities of different hybrid models adopted from prior works.

## 3 Experimental Setup

### 3.1 Task Overview

We evaluate in-context retrieval through two complementary tasks shown in Figure 1. We utilize synthetic tasks introduced in prior work (Jelassi et al., 2024; Pantazopoulos et al., 2024), which serve as tractable proxies for sequence modeling behavior at scale. Both tasks strip away semantic content, isolating the architectural properties under study rather than parametric knowledge encoded in model weights.

**N-gram retrieval**  Given an input sequence and a query $n$-gram, the model must reproduce the $k$ tokens that immediately follow that $n$-gram in the sequence. In the standard **suffix** setting, the query appears at the end of the input, targeting a form of associative recall. This requires recurrent models to maintain a representation of the entire context before the query pattern is provided. We additionally conduct experiments with a **prefix** variant, where the query is provided first, shifting the demand from recall to selective copying: the model can encode the query and discard non-matching tokens as it processes the sequence.

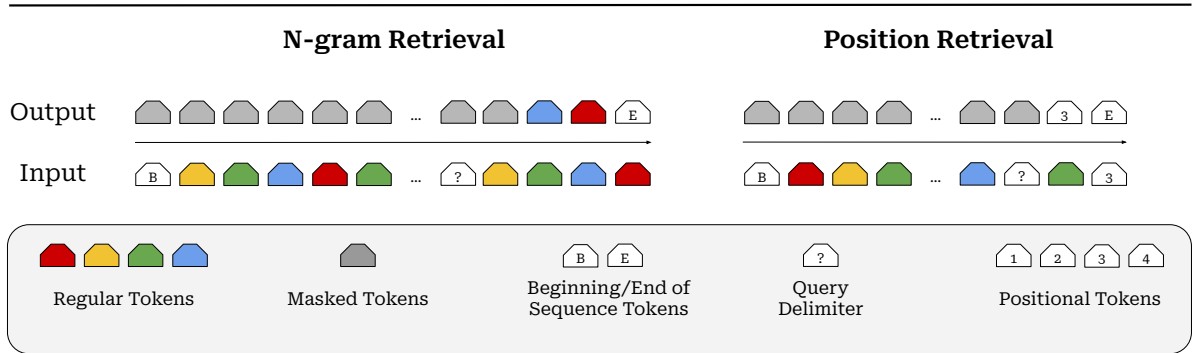

Figure 1: Examples of training sequences for the two in-context retrieval tasks. **Left:** In n-gram retrieval, the model accepts a sequence containing a query n-gram (e.g., $n = 2$, 🟡 🟢) and must produce the $k$ tokens following in the sequence (e.g., $k = 2$, 🔵 🔴). **Right:** In position retrieval, the model accepts a sequence with a single query token (🟢), and must output the positional index of that token in the sequence (here, 3). Position indices are represented as dedicated vocabulary tokens distinct from regular input tokens.

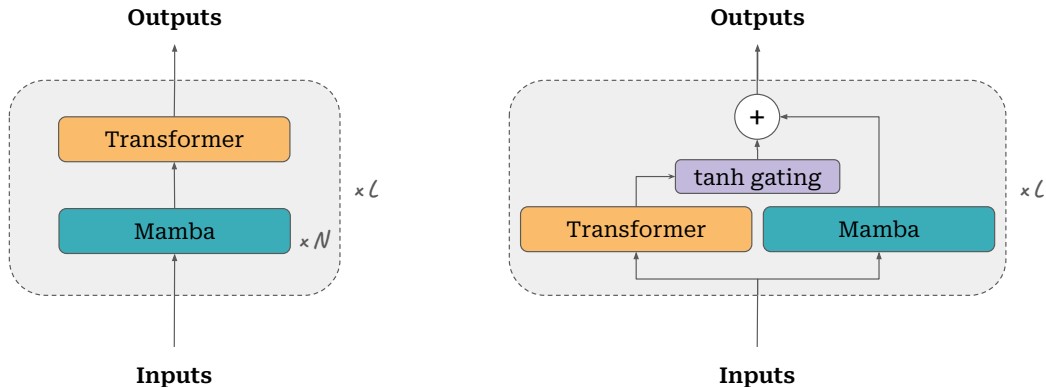

Figure 2: Illustration of hybrid architectures. **Left:** Interleaved Mamba and Transformer blocks. A Transformer block is inserted every $N$ Mamba blocks. **Right:** Two stream block with a gate mechanism. The outputs from both streams are fused with a learnable gating mechanism.

**Position retrieval** This "two-hop" associative task evaluates the ability to map a query token to a specific positional index. Given a sequence followed by a query token, the model must 1) locate that token within the sequence, and 2) output its position represented as dedicated "coordinate" tokens within the model's vocabulary. This structure mirrors a broad family of cross-modal grounding tasks, such as referring expression comprehension (Kazemzadeh et al., 2014), GUI grounding (Cheng et al., 2024), video moment retrieval (Zhang et al., 2023), and robotic manipulation (Kim et al., 2024; Wen et al., 2025a), where the challenge is not only retrieval of a referent but also translating into spatial or temporal coordinates.

## 3.2 Model architectures

We experiment with three families of decoder-only architectures: Transformers, State Space Models (SSMs), and Hybrid models combining Transformer and Mamba blocks, as shown in Figure 2. As SSM baselines, we consider both Mamba (Gu & Dao, 2023) and Mamba2 (Dao & Gu, 2024). Following prior work (Jelassi et al., 2024), we define hyperparameters such that all models are matched in parameter size ($\approx 150 - 160M$ parameters). Supplementary information regarding the configuration of each model is provided in Section A.

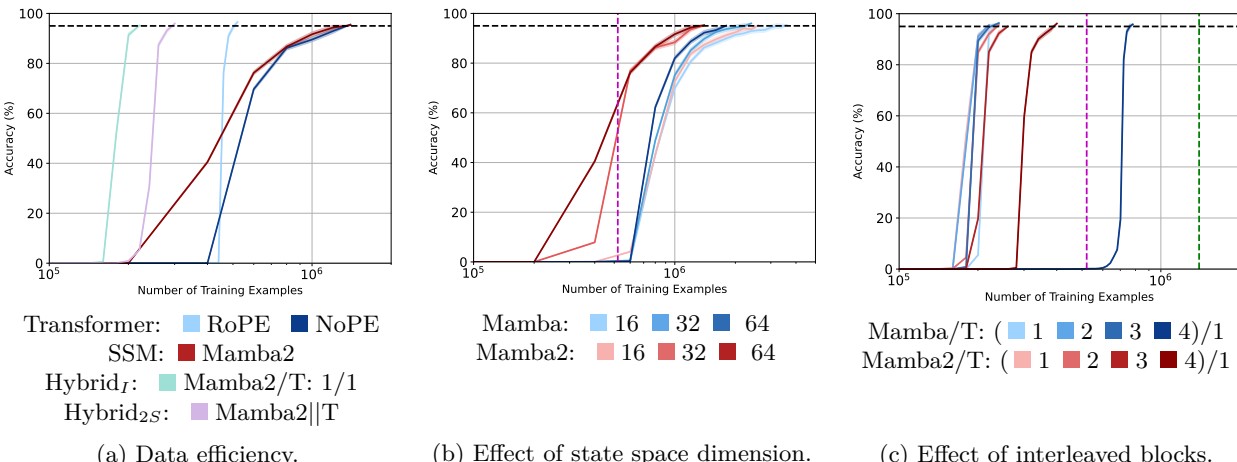

(a) Data efficiency.  (b) Effect of state space dimension.  (c) Effect of interleaved blocks.

Figure 3: **(a) N-gram retrieval data efficiency.** We train models to retrieve a sequence of $k = 3$ tokens that follow a randomly selected n-gram ($n = 2$) in a string of length $\leq 100$, and evaluate on strings of length 100. While Transformers train significantly faster than SSMs, hybrid architectures are converging even faster. **(b) Effect of state space dimension.** We train SSM models with different state space dimensions. Both Mamba versions benefit from a higher capacity state space, their performance is still inferior to that of a standard Transformer, shown here as a violet dashed line. **(c) Effect of interleaved SSM/Transformer blocks in hybrid-interleaved models (Hybrid$_I$).** We explore the effect of a Transformer block after $N = \{1, 2, 3, 4\}$ SSM blocks. A single Transformer layer complements the SSM stack by correcting the hidden state and yielding increased performance than a pure SSM model shown in green. Models with interleaving Transformer blocks after $N < 4$ SSM blocks even surpass the performance of a pure Transformer.

**Transformer positional encodings**  Rotary positional embeddings (RoPE) (Su et al., 2024) have become the standard alternative to absolute (Vaswani et al., 2017) or learned (Devlin et al., 2019) positional encodings. However, their performance still degrades on sequences exceeding the training context window (Dubois et al., 2020; Press et al., 2022). As such, many methods adapt RoPE embeddings on longer sequences with expensive long-context fine tuning (Zhu et al., 2024; Ding et al., 2024; Peng et al., 2024). An alternative line of work fully omits positional information (NoPE) (Kazemnejad et al., 2023), showcasing improvements in length generalization. We therefore evaluate both standard RoPE and NoPE-based Transformer baselines.

**Hybrid architectures**  Hybrid designs aim to address the in-context retrieval limitations of SSMs (Jelassi et al., 2024; Pantazopoulos et al., 2024). Since Transformer blocks have access to all prior tokens, these blocks may learn to edit the SSM's hidden state with information discarded during a previous timestep. We investigate two strategies for fusing Transformer and SSM layers, reflecting design choices in recent large-scale models: an interleaved setup (**Hybrid$_I$**), where a Transformer block follows every $N \in \{1,2,3,4\}$ SSM blocks (Lenz et al., 2025; Blakeman et al., 2025), and a two-stream setup (**Hybrid$_{2S}$**), where the hidden states of Transformer and SSM blocks are combined at each layer via a learnable tanh gate (Dong et al., 2025). The gate is zero-initialized such that the Transformer stream is inactive at the start of training, allowing the model to progressively learn a balance between global attention and recurrent compression. All hybrid models use RoPE for the Transformer blocks.

## 4 Experiments

### 4.1 N-gram retrieval: Learning to retrieve parts of context

We evaluate n-gram retrieval across three dimensions: data efficiency, length generalization, and robustness to duplicate queries. We consider both suffix and prefix variants (see Section 3.1), as they impose different memory demands on the model. Models are trained under the same conditions (hyperparameters, learning rate sweeps, and data seeds) on sequences of up to 100 tokens and evaluated on sequences of up to 1000

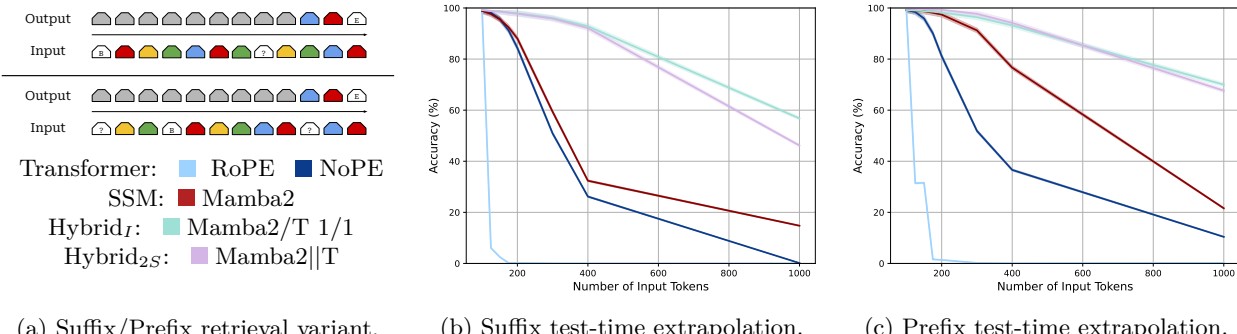

(a) Suffix/Prefix retrieval variant.     (b) Suffix test-time extrapolation.     (c) Prefix test-time extrapolation.

Figure 4: **(a)** Illustration of the suffix (top) / prefix n-gram retrieval (bottom) variants. In the suffix version the query is given at the end of the input sequence, while in the prefix version the query is provided at the beginning. **(b)** When training with sequences $\leq 100$ Mamba2 exhibits greater generalization than a Transformer with RoPE embeddings, while hybrid models show near-perfect generalization abilities. **(c)** On the prefix variant, Mamba2 performs even better given the lower memory requirements of the task, while Transformers exhibit a slight performance boost.

tokens for length generalization. For duplicate query experiments, we insert multiple instances of the query n-gram into the input sequence, each followed by a distinct k-gram. In all experiments, $n = 2$ and $k = 3$.

**Data efficiency** Figure 3a shows the data efficiency of different models on the suffix variant. Hybrid architectures, particularly those with a Mamba2 backbone, converge fastest, requiring an order of magnitude less data than standalone SSMs to reach near-perfect accuracy ($\geq 95\%$). Transformers with RoPE fall in between, while Transformers without positional encodings (NoPE) are as data-inefficient as SSMs, suggesting that positional information plays an important role in associative recall. Figure 3b further shows that increasing the SSM state space dimension improves performance for both Mamba variants, though even the highest-capacity SSMs still underperform a standard Transformer with RoPE. Finally, Figure 3c examines the effect of Transformer block frequency in hybrid interleaved Hybrid$_I$ models. Even a single Transformer block inserted after $N = 4$ SSM layers dramatically improves over a pure SSM. Moreover, models with higher Transformer frequency ($N < 4$) surpass the efficiency of a standalone Transformer, suggesting a synergistic interaction: Mamba compresses the input context into a compact hidden state, while self-attention enables precise content-based retrieval over that representation. Taken together, these results indicate that hybrid models successfully combine the complementary strengths of Transformers and SSMs, where self-attention interacts positively with the recurrent update rule of Mamba.

> ***Finding* 1.** Hybrid architectures are more data-efficient compared to standalone Transformers or SSMs on n-gram retrieval, with interleaved models achieving the best performance.

**Length extrapolation** Next, we focus on length generalization capabilities. We train all models with examples of at most 100 tokens until they achieve perfect accuracy and evaluate on sequences up to 1000 tokens. The results are presented in Figure 4. Overall, all models exhibit a performance boost on the prefix variant, which is more evident in the case of Mamba2. This behavior is expected particularly for SSMs given the low memory requirements of the prefix variation. Transformers with standard positional embeddings exhibit poor generalization, a finding confirmed by many prior works (Hupkes et al., 2020; Newman et al., 2020; Dubois et al., 2020; Lee et al., 2025). Transformers with NoPE, however, behave similar to Mamba2 on the suffix variant. Perhaps most interestingly, both hybrid variants score very high in terms of generalization and only exhibit performance degradation on $10\times$ longer sequences. The attention mechanism on top of the hidden state of an SSM provides complementary information, by essentially correcting the hidden state at each time step and enabling length generalization at least within the sequence length that we have experimented with. We note however, that the hybrid models shown in Figure 4 contain the highest

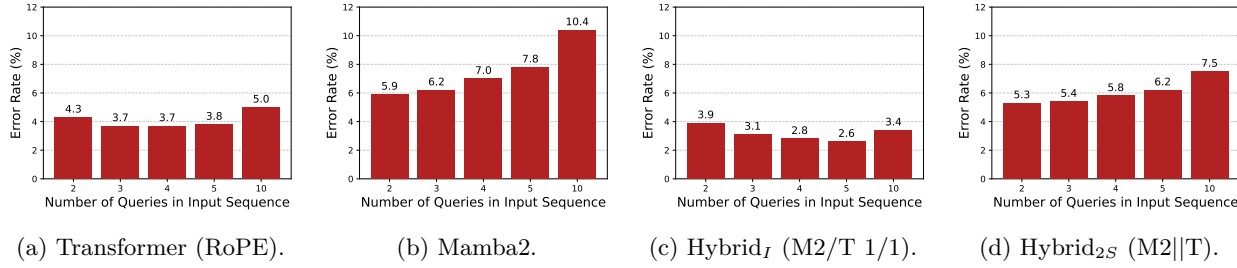

Figure 5: Error rates with non unique n-gram suffix queries of a model from each family: **(a)** Transformer, **(b)** SSM: Mamba2, **(c)**: Hybrid$_I$ and, **(d)** Hybrid$_{2S}$ with interleaved or parallel Mamba2 and Transformer blocks. Mamba2 fails to match the query n-gram to any candidate within the sequence, while a Transformer and the hybrid models maintain low miss rate even for sequences with many duplicates.

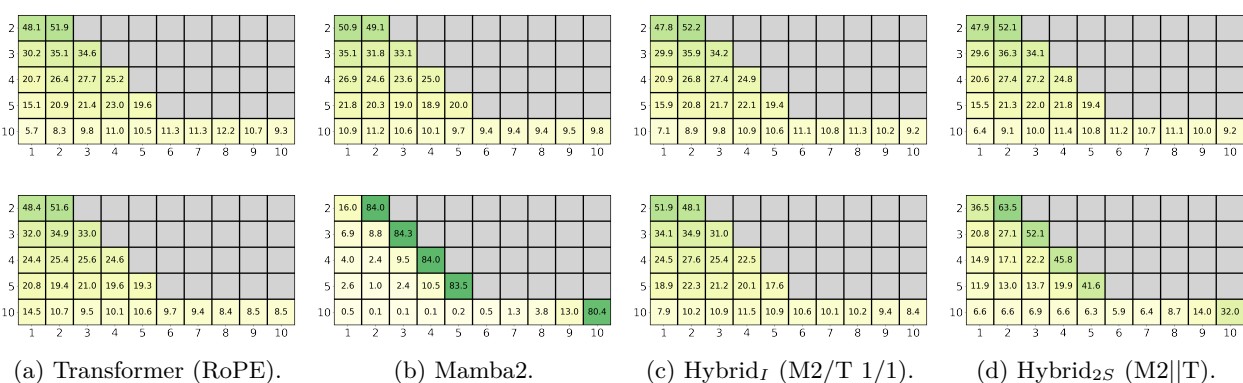

Figure 6: Preference rates with non unique n-gram suffix (top) and prefix (bottom) queries across different bins within sequences for each model: **(a)** Transformer, **(b)** SSM: Mamba2, **(c)**: Hybrid$_I$ and, **(d)** Hybrid$_{2S}$ with interleaved or parallel Mamba2 and Transformer blocks. Each row in the heatmap corresponds to dividing the sequence into $s = \{2, 3, 4, 10\}$ segments containing duplicate queries.

number of Transformers layers. We provide ablations regarding the number of Transformers layers for length extrapolation in Figure 12.

> **_Finding_ 2.** Hybrid models generalize to sequences longer than seen during training, outperforming both standalone Transformers (which degrade with RoPE) and standalone SSMs (which generalize only in lightweight memory settings).

**Duplicate queries in input sequences** In the previous experiments, we assume that the query is unique within the sequence. We investigate a more adversarial scenario in which, during evaluation, the input sequence contains duplicate entries of the query n-gram. For this purpose, we divide the sequence into equally sized segments, and insert duplicates of the sampled n-gram query into each segment, ensuring that the expected k-gram is different in each duplicate. For example, in a sequence containing three occurrences of the query n-gram, each instance is randomly positioned within the first, second, and third segment of the input sequence, respectively. Similarly to test-time extrapolation, we train all models with examples of at most 100 tokens until they reach a perfect score. For evaluation, we compute the preference and error rates for each model when trying to match the predicted k-gram to any of the ground-truth k-grams associated with the duplicate query instances in the sequence.

Figure 5 presents the average error rates across different model families for the prefix variant showing that the hybrid interleaved model achieves the lowest error rates across all models, even with 10 duplicate queries, a scenario where, for $n = 2$ and $k = 3$ the duplicate queries constitute 50% of the total sequence tokens.

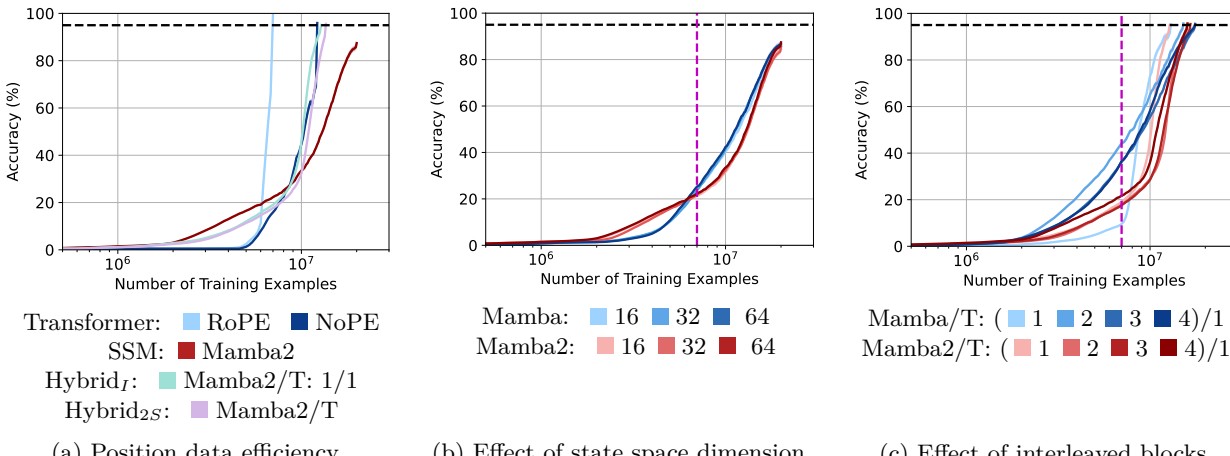

(a) Position data efficiency.   (b) Effect of state space dimension.   (c) Effect of interleaved blocks.

Figure 7: **(a) Position retrieval data efficiency.** We train models to retrieve the position of a token randomly sampled from a sequence of 200 tokens. Transformers converge fastest, while hybrid architectures require more steps and SSMs fail to learn the task entirely within the training budget. **(b) Effect of state space dimension.** Even SSMs with the largest state space dimensions are inferior compared to a standard Transformer, shown here as a violet dashed line. **(c) Effect of interleaved SSM/Transformer blocks in hybrid-interleaved models (Hybrid$_I$).** A higher density of Transformer blocks speeds up optimization and approximates the performance of pure Transformer.

We further analyze the performance by examining which of the repeated queries within the segments was selected by each model in Figure 6. Importantly, we observe that Transformers and Hybrid$_I$ models do not form positional biases as they uniformly select candidate k-grams from each segment both for the suffix and the prefix version of the task. In contrast, Mamba2 exhibits strong positional bias when presented with the query at the beginning of the sequence, as in more than 80% of the cases, it selects the k-gram belonging to the last segment. This behavior is also shown to a lesser degree in the case of the two-stream hybrid model. Overall, these results further reinforce the effect of the task formulation also shown during the test-time extrapolation experiments. Ultimately, we would like architectures that are robust to both task formulation and input perturbations, maintaining consistent retrieval performance regardless of query placement or the presence of spurious duplicate tokens. Our results suggest that hybrid interleaved models represent a promising step in this direction, combining the strengths of both Transformers and SSMs while mitigating their respective positional biases.

> **_Finding_ 3.** When input sequences contain duplicate query n-grams, the interleaved hybrid model achieves the lowest error rates across all architectures and avoids the strong recency bias exhibited by Mamba2, which in the prefix setting selects the final-segment k-gram in over 80% of cases.

## 4.2 Position retrieval: Learning to retrieve position of elements in context

We now explore the task of retrieving the position of an element in the sequence. Unlike the n-gram retrieval task, the model needs to perform a two-hop association by first identifying the correct token in the input sequence and then mapping it to the corresponding positional vocabulary token. We evaluate data efficiency and learning dynamics on sequences of 200 tokens, and further analyze the internal representations learned by each architecture to explain the observed performance differences. Models are trained until reaching 95% validation accuracy or exhausting a budget of 20M training examples.

**Data efficiency** Results are presented in Figure 7. A Transformer with RoPE embeddings converges faster than other models. In contrast to n-gram retrieval, Transformers with RoPE now converge fastest across all architectures, while hybrid models, despite their advantage on n-gram retrieval, learn the task at the

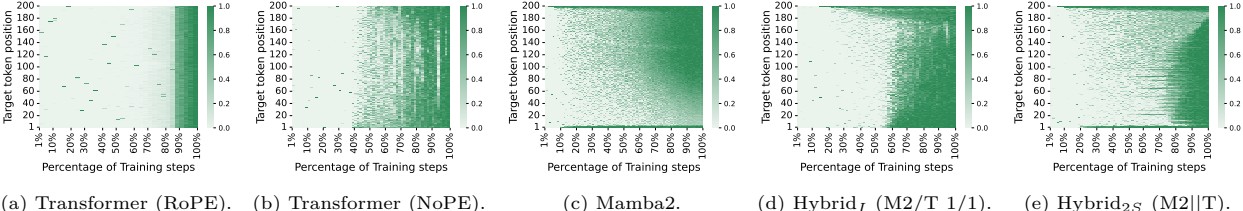

| (a) Transformer (RoPE). | (b) Transformer (NoPE). | (c) Mamba2. | (d) Hybrid$_I$ (M2/T 1/1). | (e) Hybrid$_{2S}$ (M2||T). |
|---|---|---|---|---|

Figure 8: We track the per-token performance of a model from each family. A Transformer model learns the task faster and with a uniform per-target-token performance distribution. Models with SSM blocks prioritize the performance at the beginning and end of the sequence, then gradually learn the task for the remaining positions. Hybrid models combine the prioritization of SSM models at the early training stages with the steep uniform distribution of Transformers.

same rate as NoPE Transformers, requiring approximately 12M examples to reach the accuracy threshold ($\approx$ 5M more than RoPE). Meanwhile, standalone SSM models fail to reach the accuracy threshold within the training budget, regardless of the state space capacity (Figure 7b). Finally, we observe that increasing the Transformer block frequency in Hybrid$_I$ models narrows the gap with standard Transformers (Figure 7c), suggesting that denser attention is beneficial for the position retrieval task.

> **_Finding_ 4.** Transformers learn to solve the position retrieval task faster than any other model. Hybrid models approximate the performance of the Transformer with more self attention blocks but standalone SSMs are not able to solve the task within the training budget.

**Learning dynamics** The analysis of model performance as a function of training set size revealed that SSMs and hybrid architectures exhibit faster initial task acquisition compared to Transformer models. This was evident to some degree in Figure 3a, but it is clearly shown in Figure 7a, where models with SSM blocks have non-negligible accuracy scores significantly earlier in the training compared to Transformers. This phenomenon is further illustrated through training loss dynamics in Figure 17. Models employing SSM backbones demonstrate the ability to identify plausible solutions after approximately 2M training examples, subsequently exhibiting gradual improvement in task performance. In contrast, the Transformer model exhibits a markedly different learning trajectory, characterized by a steep decline in loss only after exposure to 7M examples. These observations motivate our investigation into the underlying factors contributing to this performance disparity. For this purpose, we track the per-target-token performance of all models, i.e., the accuracy of a model when the query refers to the $l$-th token in the sequence ($1 \leq l \leq 200$). The results are illustrated in Figure 8. A Transformer model solves the task almost instantaneously regardless of the position of the query token, which is explained by the training loss curve. However, SSM models behave quite differently. We observe that even from 10% of the total training steps, these models can solve the task when the query refers to the head or the tail of the sequence, while, during the same timestep, the Transformer does not. Additionally, these models learn to gradually solve the task for intermediate positions in the sequence compared to the Transformer. This means that Mamba models choose to retain information about the first and last tokens in the sequence, a phenomenon observed in humans and commonly referred to as the serial position effect (Murdock Jr, 1962), where items at the beginning and at the end of a list are recalled better than those in the middle. Finally, we note that hybrid models exhibit a similar trend, where from the early stages of the training, they can solve the task for queries at the end of the sequence, but also show a steep performance increase as the Transformer. With regards to interleaved hybrid models, increasing the frequency of the attention block encourages more steep behavior as shown in Figure 13.

> **_Finding_ 5.** SSMs and hybrid models acquire early partial solutions by prioritizing tokens at the beginning and end of the sequence, while Transformers remain near-random until a sharp, uniform performance transition occurs, solving the task for all positions in the sequence.

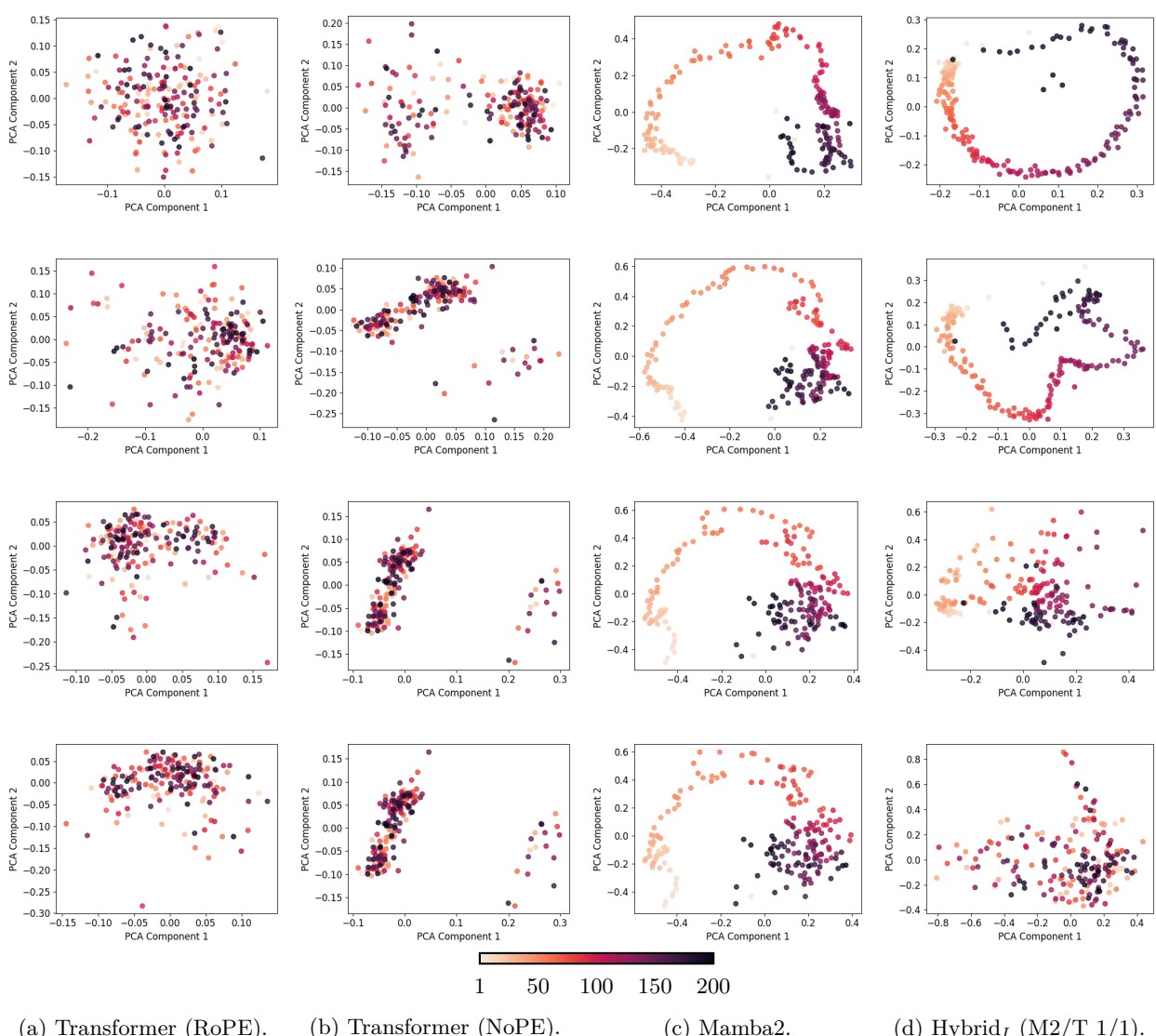

(a) Transformer (RoPE).  (b) Transformer (NoPE).  (c) Mamba2.  (d) Hybrid$_I$ (M2/T 1/1).

Figure 9: We track the embeddings of the position tokens in a 2D plane across training for **(a)** Transformer with RoPE positional embeddings **(b)** Transformer without positional embeddings, **(c)** Mamba2, and **(d)** Hybrid with interleaved Mamba2 and Transformer blocks. Each row corresponds to 25% of the training progress relative to each model, while warmer colors in the spectrum correspond to positional tokens pointing to the start of the sequence. SSMs learn a smooth, continuous mapping of positions by converging to locality-aware embeddings that form a two-dimensional spiral. Transformers learn a mapping between the positional token and the actual position in the sequence without any particular structure.

### 4.2.1 SSMs develop locality-aware representations

Previously, we showed that Transformers learn to map tokens to positions regardless of their index within the sequence, while SSMs prioritize solving the task concerning tokens at the beginning and at the end of the sequence. Consequently, we analyze the representations of these models by inspecting the embeddings of the position tokens. Recall that if $i$ refers to the position of the query token in the sequence, we expect the model to output the token $p_i$. Since the examples are randomly sampled, there is no correlation between $p_i$ and the value of the $i$-th token in the sequence. As such, any difference between the representation of the tokens $p_{i-1}$, $p_i$, $p_{i+1}$ that describe adjacent positions, must be attributed to the model's capacity of differentiating between the indices $i-1, i, i+1$.

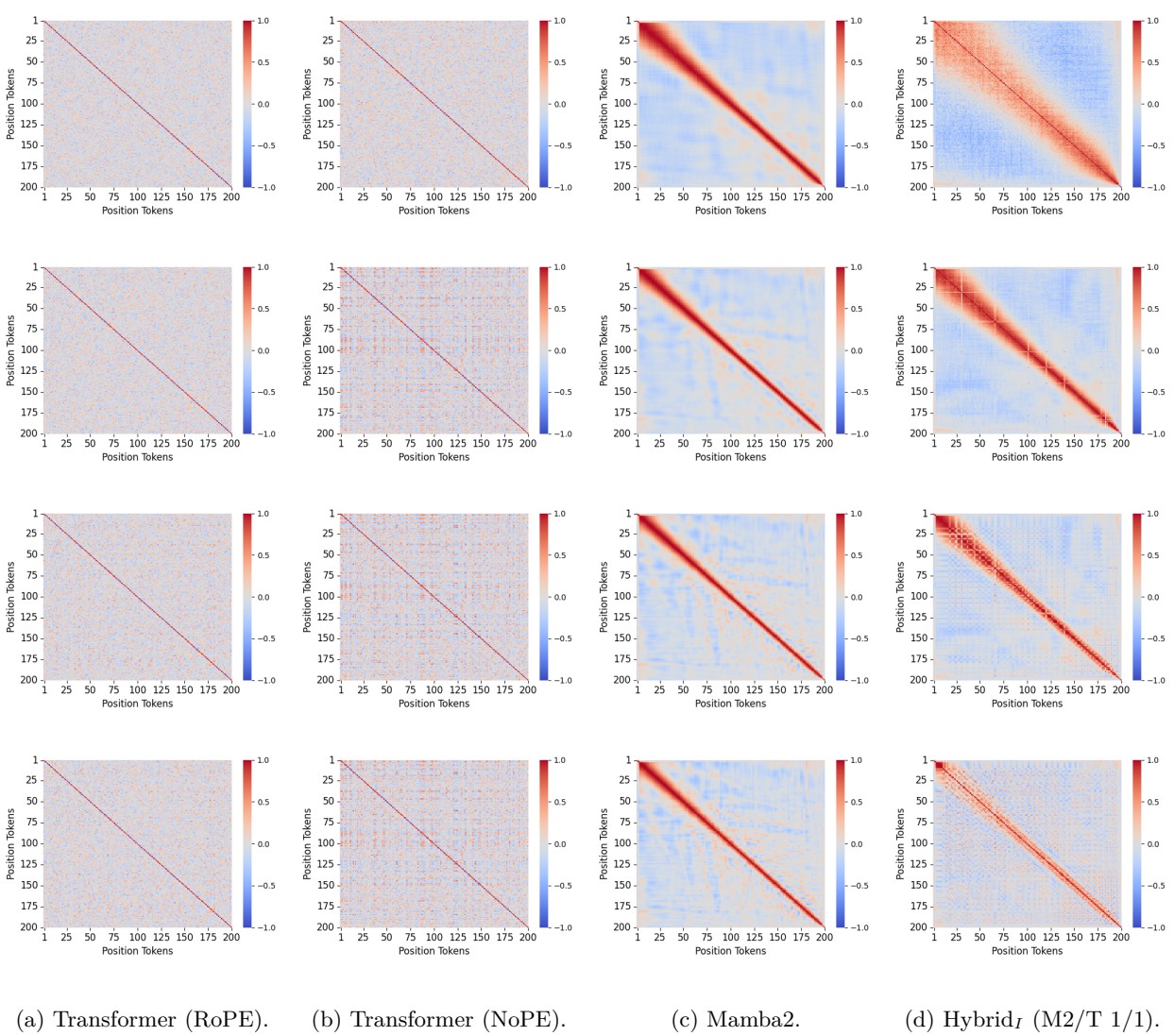

(a) Transformer (RoPE).      (b) Transformer (NoPE).      (c) Mamba2.      (d) Hybrid$_I$ (M2/T 1/1).

Figure 10: We track the cosine similarities between embeddings of position tokens projected using PCA (dim=32). **(a)** Transformer with RoPE positional embeddings **(b)** Transformer without positional embeddings, **(c)** Mamba2, and **(d)** Hybrid with interleaved Mamba2 and Transformer blocks. Each row corresponds to 25% of the training progress relative to each model. Models with SSM blocks produce similar embeddings for tokens that describe nearby indices in the sequence as elements near the main diagonal have very high similarity scores. Transformers do not have this property as they learn a mapping between inputs-outputs without taking into account adjacent positions.

For this purpose, we begin by visualizing the embeddings of the position tokens every 25% of the training using PCA. The results are illustrated in Figure 9. Surprisingly, we observe that SSM-based models form structured latent representations that represent position simply via next-token prediction and without any task-specific supervision[2]. Specifically, these representations form a spiral within the embedding space which can be tracked into a 2D-dimensional space that preserves locality, i.e., tokens that represent neighbor positions in the sequence are neighbors within the spiral trajectory. In fact, this behavior is incentivized from early training stages and fully preserved throughout the training for pure SSM models, while hybrid architectures deviate from the spiral-like structure as the training progresses. However, all of our models that

---

[2]For instance, a regularization term $|i - j|$ penalizing the model after predicting the index $j$ proportionally to the distance from the ground truth index $i$.

adopt at least one SSM block converge to the same neighborhood structure; 1) tokens that depict adjacent positions in the sequence are neighbors within the embedding space, and 2) positional information is encoded in a low-dimensionality subspace of the embedding space (see Figure 10 and Section B.5). Additionally, we also observe that the locality property is initially developed in tokens depicting beginning and end positions in the sequence, which further corroborates the learning dynamics shown in Figure 8. Importantly, this is not the case in Transformers, where the embeddings of these tokens are densely cluttered, implying a simple mapping between position-outputs without a locality structure. We hypothesized that this is due to the RoPE embeddings providing the necessary positional information, but even a Transformer without any positional information does not develop this property. Finally, we further highlight this behavior in Figure 11, where we show the average absolute distance between every position $i$ and its $K$ nearest neighbors for the Transformer and Hybrid$_I$. Notably, the hybrid model maintains significantly lower distances between neighbors compared to the Transformer even for high values of $K$.

Consequently, we attribute this behavior to the update rule of the hidden state of each model. In principle, the Transformer can look at all tokens in the past to update the hidden state at each time step. SSMs and Mamba in particular force a very unique and strict update of the hidden state which is based solely on the state from the previous time step and the current input. As such, Mamba performs local updates that converge to this structure. Regardless, while one would expect that the representations of adjacent positions to be neighbors, this property is not necessary to solve the underlying task but may in fact function as a bottleneck regarding efficiency, which justifies the slow convergence of hybrid models compared to the results presented in Section 4.1.

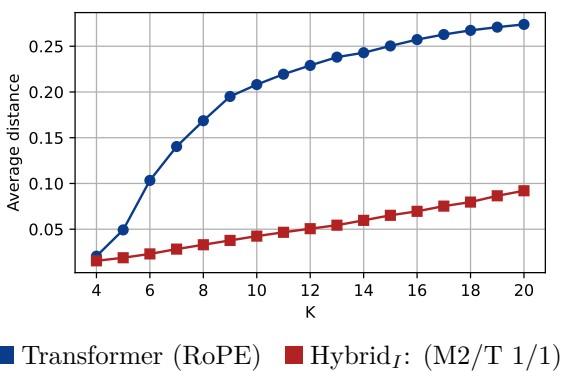

Figure 11: Average absolute distance of K-nearest neighbors of position embeddings.

> ***Finding* 6.** SSMs learn locality-preserving positional embeddings that form a two-dimensional spiral, keeping adjacent positions as neighbors. Transformers develop no such structure, with or without positional encodings. Hybrid models, similar to SSMs, develop locality-preserving embeddings, where adjacent positions remain neighbors and positional information occupies a low-dimensional subspace.

## 5  Conclusion

This work provides a systematic analysis of in-context retrieval capabilities across Transformers, SSMs, and hybrid architectures under controlled experimental conditions. Using two complementary synthetic tasks, n-gram retrieval and position retrieval, we identified nuanced trade-offs in how these architectural families learn to retrieve information from context. Hybrid models outperform both pure Transformers and SSMs on n-gram retrieval in terms of data efficiency, length generalization, and robustness to duplicate queries. For position retrieval, Transformers maintain the lead while models with SSM blocks lag behind. We attribute this to the locality-aware embeddings that SSM blocks induce, which can be considered as an emergent and interpretable property, but one that ultimately hinders precise two-hop associative lookup. Rather than treating SSM and attention blocks as disjoint components, their complementary inductive biases may be exploited for more robust long-context modeling towards the next generation of sequence modeling networks.

**Limitations**  Our study focuses on synthetic tasks with controlled complexity, showing important aspects of Transformers, SSMs, and hybrid architectures regarding in-context retrieval capabilities. While these tasks correlate with practical capabilities, they may not fully capture all aspects of real-world sequence modeling. As such validation on full-scale language modeling and multimodal tasks remains important future work. Finally, our analysis focuses on decoder-only architectures. The interplay between SSMs and attention in encoder-decoder or encoder-only settings may exhibit different characteristics and warrants separate investigation.

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

# A   Experimental Setup

## A.1   Synthetic Tasks

**N-gram retrieval**   All experiments are conducted using a vocabulary of 30, the size of the query n-gram is $n = 2$ and the output k-gram is $k = 3$. Additionally, all sequences during training and evaluation are randomly sampled from the vocabulary with replacement ensuring that the query n-gram appears only once within the sequence. Note that this does not prevent the existence of other unique n-grams. With regards to the duplicate experiments, we ensured that the query appears multiple times and each duplicate is followed by a unique k-gram.

**Position retrieval**   All experiments are conducted using a vocabulary of 200 tokens describing the input sequence and an additional 200 tokens for their position. Each sequence during training and evaluation is constructed by randomly shuffling the input 200 tokens.

## A.2   Model architecture

Table 1 shows the configuration of all models. For the Transformer blocks used the GPT-NeoX archi-tecture[3]. Regarding the Mamba models we consider three variants with different state space dimensions $S \in \{16, 32, 64\}$, while preserving a similar number of parameters. Finally, for hybrid models we set $S = 16$ across all experiments. Notably, we expect that, similar to the findings in Sections 4.1 and 4.2, using a higher dimension for the state space will likely result in greater performance. However, our goal in the experiments was to illustrate the impact of self-attention on correcting the hidden state of an SSM. We therefore compared the highest-capacity SSM models with the minimum-capacity hybrid models. Similarly, we opted for RoPE embeddings for all Transformer blocks within the hybrid models as opposed to omitting any positional information.

**Software dependencies**   All experiments were conducted using Hugging Face (Wolf et al., 2020) (`v4.57.3`), and the Mamba repository (Gu & Dao, 2023).

## A.3   Model training

**Training hyperparameters**   Table 2 shows the training hyperparameters used during training. Additionally, all models are approximately computed-matched requiring $\approx 30 \times 10^9$ FLOPs for a single forward pass assuming an input sequence of 100 tokens which corresponds to an example from the n-gram retrieval task, empirically estimated using standard practices[4] (Kaplan et al., 2020). We train all models under identical settings using three different random seeds for data sampling and model initialization. We performed three sweeps for each data seed resulting in 9 different runs per model. We train all models for next-token predic-tion and only penalize the model for mistakes on the expected response and not on the inputs of an example. All experiments were conducted on a single NVIDIA H100 80GB HBM3.

**Evaluation**   With regards to evaluation we use the string-level accuracy across both n-gram and position retrieval tasks. In particular, for the n-gram retrieval a correct prediction corresponds to predicting exactly the $k$ characters under teaching forcing. For measuring the test-time extrapolation (Section 4.1), we apply greedy decoding by setting the temperature scaling to 0.

---

[3]`https://github.com/huggingface/transformers/blob/main/src/transformers/models/gpt_neox/modeling_gpt_neox.py`

[4]`https://github.com/state-spaces/mamba/issues/110`

| | Params (M) | Layers | Model Dim | Attn Heads | Pos Emb | SSM Dim | Gate $\alpha$ |
|---|---|---|---|---|---|---|---|
| **Transformer** | 151 | 12 | 1024 | 16 | RoPE / NoPE | - | - |
| **Mamba** | 160 | 24 | 1024 | - | - | 16 | - |
| **Mamba** | 162 | 24 | 1024 | - | - | 32 | - |
| **Mamba** | 167 | 24 | 1024 | - | - | 64 | - |
| **Mamba2** | 152 | 24 | 1024 | - | - | 16 | - |
| **Mamba2** | 153 | 24 | 1024 | - | - | 32 | - |
| **Mamba2** | 155 | 24 | 1024 | - | - | 64 | - |
| **Hybrid$_I$ 1M/1T** | 154 | 16 | 1024 | 16 | RoPE | 16 | - |
| **Hybrid$_I$ 2M/1T** | 155 | 18 | 1024 | 16 | RoPE | 16 | - |
| **Hybrid$_I$ 3M/1T** | 162 | 20 | 1024 | 16 | RoPE | 16 | - |
| **Hybrid$_I$ 4M/1T** | 157 | 20 | 1024 | 16 | RoPE | 16 | - |
| **Hybrid$_{2S}$ M‖T** | 154 | 8 | 1024 | 16 | RoPE | 16 | 0 |
| **Hybrid$_I$ 1M2/1T** | 151 | 16 | 1024 | 16 | RoPE | 16 | - |
| **Hybrid$_I$ 2M2/1T** | 152 | 18 | 1024 | 16 | RoPE | 16 | - |
| **Hybrid$_I$ 3M2/1T** | 158 | 20 | 1024 | 16 | RoPE | 16 | - |
| **Hybrid$_I$ 4M2/1T** | 152 | 20 | 1024 | 16 | RoPE | 16 | - |
| **Hybrid$_{2S}$ M2‖T** | 151 | 8 | 1024 | 16 | RoPE | 16 | 0 |

Table 1: Model configuration. We consider models with ≈150-160M parameters. Hybrid models with interleaved Transformer and Mamba blocks are denoted as $NM(2)/1T$ where $N$ denotes the number of Mamba (M) or Mamba2 (M2) blocks preceding a Transformer block. Models with a two-stream hybrid block are denoted as ‖, where the output of the two streams is combined via a learnable gated tanh mechanism. The gate is a scalar value initialized to zero, meaning that the Transformer stream is inactive at the start of the training. All parameters are computed excluding the embedding table as the vocabulary size differs between the two tasks.

| Hyperparameter | Values |
|---|---|
| Training Examples | $20 \times 10^6$ |
| Evaluation Examples | $10^5$ |
| Seeds | {12345, 123456, 1234567} |
| Global batch size | 64 |
| Learning rate | {1e-05, 5e-05, 1e-04} |
| Learning rate schedule | cosine decay |
| Learning rate warmup | 0.1 |
| Number of training steps | 312500 |
| Optimizer | AdamW |
| Adam beta1 | 0.9 |
| Adam beta2 | 0.999 |
| Adam epsilon | 1e-08 |
| Weight decay | 0.01 |
| Max grad norm | 1.0 |

Table 2: Training hyperparameters used across all models and tasks. All experiments were conducted using the Hugging Face library (Wolf et al., 2020) (`v4.57.3`).

## B  Experiments

### B.1  Ratio of Transformers/SSMs in interleaved hybrid models

**Test-time extrapolation**  Figure 12 illustrates the test-time extrapolation of different hybrid interleaved (Hybrid$_I$) configurations. Overall, we observe strong generalization capabilities across both n-gram task variants. With regards to comparisons between Mamba and Mamba2, the latter results in greater extrapolation

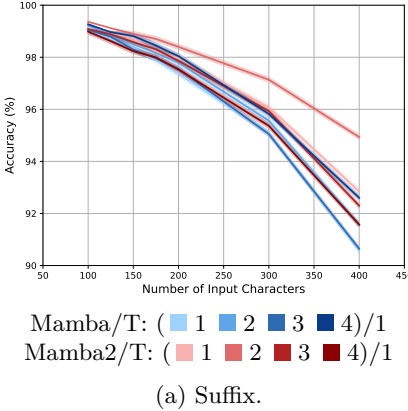

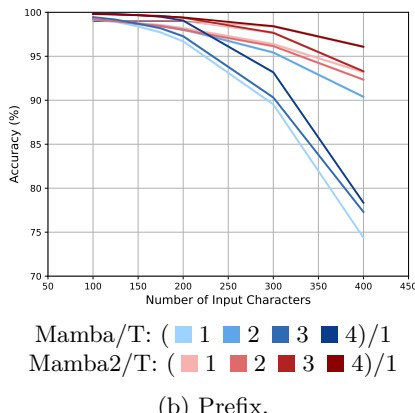

(a) Suffix.

(b) Prefix.

Figure 12: Test-time extrapolation of different hybrid interleaved configurations for the **(a)** suffix, and **(b)** prefix n-gram retrieval tasks. Hybrid models exhibit longer length generalization capabilities compared to Transformers and SSMs. The best configuration corresponds to hybrid models with Mamba2 and low frequency Transformer blocks.

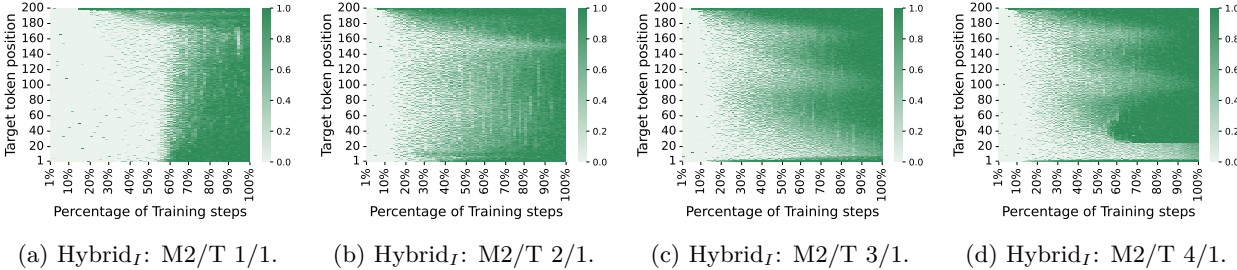

(a) Hybrid$_I$: M2/T 1/1.  (b) Hybrid$_I$: M2/T 2/1.  (c) Hybrid$_I$: M2/T 3/1.  (d) Hybrid$_I$: M2/T 4/1.

Figure 13: We track the per-token performance of different hybrid interleaved configurations. All hybrid models tend to combine the start/end of sequence tokens of SSMs and the steep behavior of Transformers. Interchanging Mamba2 and Transformer blocks ($N$=1) results in more steep task acquisition, while for $N > 1$ hybrid models behave more like pure SSMs.

capabilities. Additionally, increasing the number of Transformer blocks results in performance degradation which is more evident in the case of the suffix version. These results are in line with the findings presented in Section 4.1 showing the performance curves of Transformer (RoPE) and Mamba2.

**Position retrieval**  Figure 13 shows the per-token performance of different interleaved models throughout training. As already mentioned, hybrid models tend to combine the prioritization of SSMs at early stages with the steep uniform per-token performance of Transformers. In particular, we observe that Hybrid$_I$ models behave similar to a Transformer (RoPE) in the case of $N = 1$ (also shown Figure 8d), while for $N > 1$ they act similar to pure SSM models.

## B.2 Duplicate queries in input sequences

**Error rates in suffix/prefix variants**  For completeness, we report the error rates for the suffix/prefix versions of all models in Figures 14 and 15. Across all models, we observe consistently higher error rates in the prefix version with the exception of the hybrid interleaved model.

In the prefix version, by the time the model needs to retrieve the n-gram following the query, it has already processed multiple prior instances of the same query in its attention in the case of Transformers, or in the hidden state in the case of SSMs. For Transformers specifically, the softmax over keys creates competition between these positions at the time of retrieval but also when updating the representations for the input

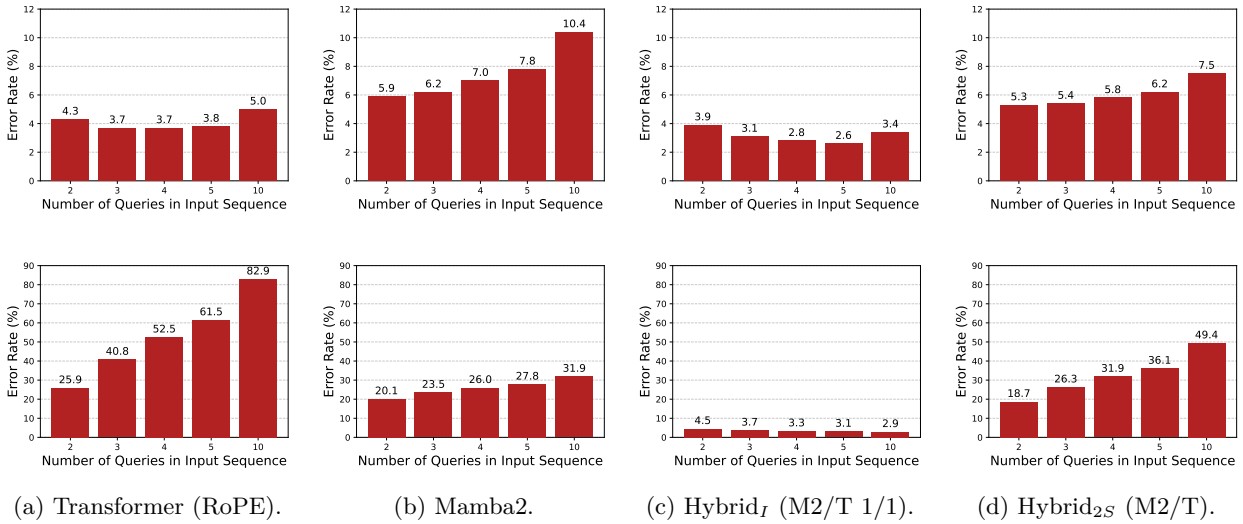

Figure 14: Error rates with non unique n-gram suffix (top) and prefix (bottom) queries of a model from each family. All models exhibit substantially higher error rates in the prefix n-gram variant apart from $\text{Hybrid}_I$.

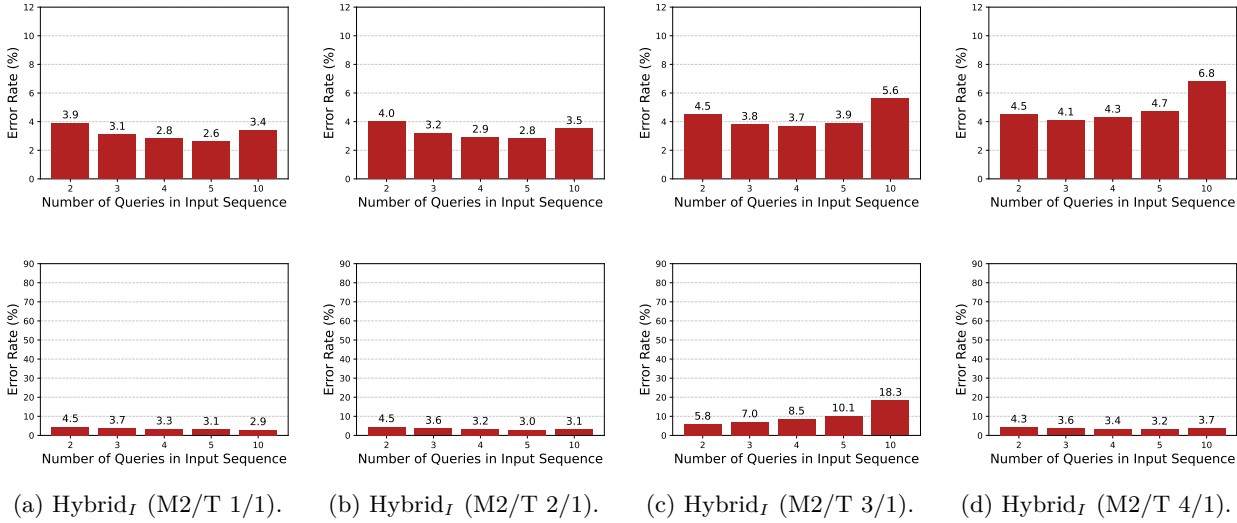

Figure 15: Error rates with non unique n-gram suffix queries of each hybrid interleaved model. All variants have significantly lower error rates compared to Transformers, and Mamba2 models (Figure 14).

sequence. Consequently, the model cannot cleanly identify which duplicate instance to anchor to for retrieval. In the suffix case the softmax introduces competition between the duplicate queries only at the time of retrieval, which is much closer to the training condition and so the error rate stays low. A similar behavior could also explain the high error rates in the case of SSMs, where the duplicates compete for the hidden state, though as Figure 6 shows, Mamba models tend to heavily prioritize the last duplicate occurrence of the query.

**Robustness of $\text{Hybrid}_I$ models** In Figure 6 we demonstrated that, similar to the Transformer, the $\text{Hybrid}_I$: Mamba2/T 1/1 model does not form positional biases with regards to selecting the candidate k-gram out of multiple duplicate queries in the sequence. Similar observations can be made for all other hybrid configurations shown in Figure 16.

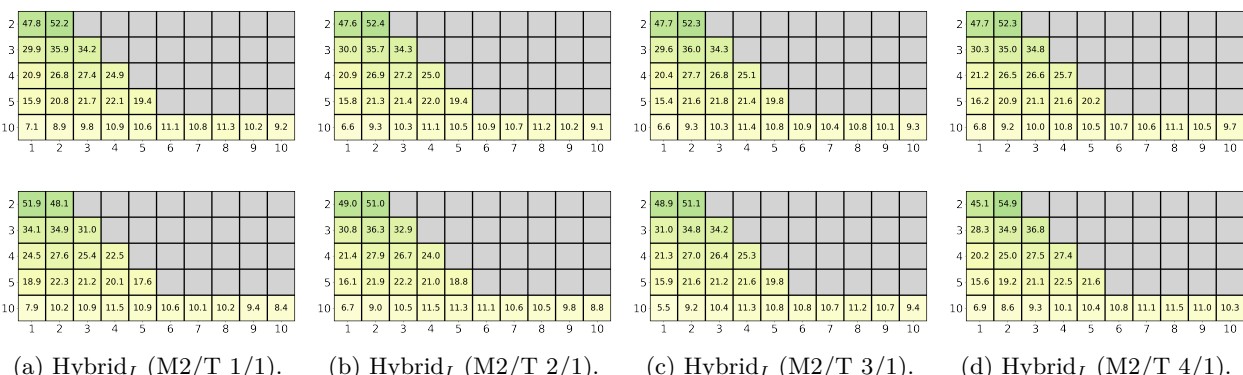

(a) Hybrid$_I$ (M2/T 1/1).     (b) Hybrid$_I$ (M2/T 2/1).     (c) Hybrid$_I$ (M2/T 3/1).     (d) Hybrid$_I$ (M2/T 4/1).

Figure 16: Preference rates with non unique n-gram suffix (top) and prefix (bottom) queries across different bins within sequences for each hybrid interleaved. Each row in the heatmap corresponds to dividing the sequence into $s = \{2, 3, 4, 10\}$ segments containing duplicate queries.

### B.3   Training loss curves on position retrieval

In Section 4.2 we explored the differences in the learning dynamics between Transformers, SSMs, and hybrid models showcasing that models containing SSM blocks begin to learn faster than Transformers. Here we further show this by formalizing the expected output distribution and examining the training loss curves.

We first note that in the position retrieval the labels correspond to two tokens, the one that indexes the query and the token that depicts the end of sequence (Figure 1). Assuming a vocabulary of $V$ tokens the expected loss for a single example would be $L = -\frac{1}{2} \sum_{i=1}^{2} l_i$, where $l_i$ is the corresponds to the log-likelihood over $V$ tokens. However, the end of sequence token is present in all training examples and as such the probability mass will concentrate over the end of sequence token. As a result, $l_2 \approx 0$ and so we are interested in the value of $l_1 = p_1 log(p_1)$. Subsequently, during training any model that has not learned anything regarding the underlying task will assign equal probability to all tokens $1/N$ and so loss value approximates:

$$l_1 = -\sum_{i=1}^{V} p(x_i) \log(p(x_i)) = -\sum_{i=1}^{V} \frac{1}{V} \log\left(\frac{1}{V}\right) = -V \cdot \frac{1}{V} \cdot \log\left(\frac{1}{V}\right) = -\log\left(\frac{1}{V}\right) = \log(V) \qquad (1)$$

Finally, for $V = 200$ the loss value for a model without knowledge of the task will be $L \approx -log(V)/2 = 2.65$, while a model that learned something about the task will express a less uniform probability distribution which results in lower loss values. Taken together, we show the learning dynamics of all models in terms of the training loss shown in Figure 17, where the horizontal gray line depicts the 2.65 threshold value. We observe that the above formalization aligns with the practical behavior of all models. In particular, all curves reach a steady state at $\approx 2.65$. After approximately 2M examples we observe that SSM models learn meaningful probability distributions resulting in loss values lower than 2.65 and gradually learn to solve the task. Transformers on the other hand have a very steep behavior where they maintain high levels of uncertainty for approximately 7M examples but then proceed to solve the task instantaneously.

### B.4   Effect of gating mechanism

**Impact of gate throughout training**   Figure 18 illustrates the magnitude of the gating mechanism in the two-stream hybrid model throughout the training process. We observe that the gate activates across all layers during the middle phase of training; however, it remains persistently active only in deeper layers for the duration of training. It is important to note that the gating mechanism employed here is not input-dependent, but rather consists of a simple scalar parameter per layer. Consequently, we anticipate that the temporal behavior and magnitude of these parameters may vary depending on initialization schemes, training regimes, and hyperparameter configurations. Nevertheless, across our experimental settings, the

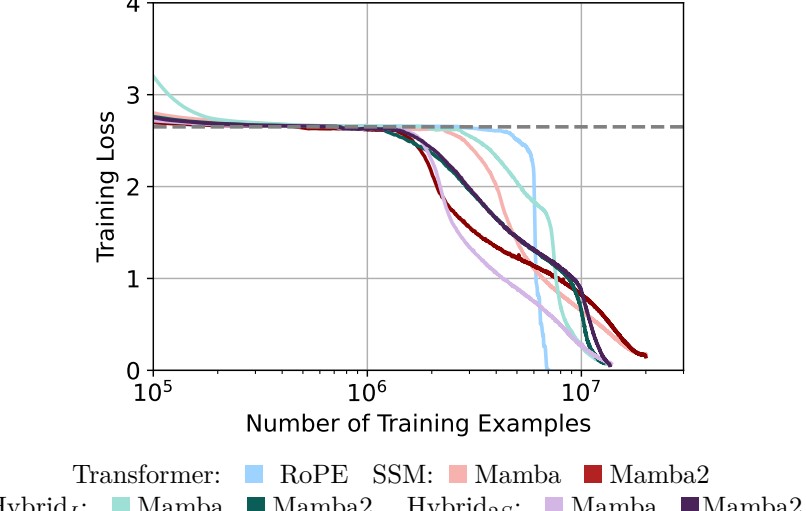

Figure 17: We track the training loss of Transformers, SSMs and Hybrid models for the position retrieval task. We observe that for the vast majority of the training Transformers remain uncertain, since they assign equal probabilities to all tokens in the vocabulary resulting in flat loss curves. Models with SSM backbones begin to learn about the underlying task significantly faster than the Transformer (albeit their slow convergence (Figure 7a compared to Transformers). Any model with loss below the horizontal value has learned something meaningful about the task.

gating effect is substantial in deeper layers, suggesting that the model learns a late fusion scheme between the two streams. This finding indicates that lower layers may process the streams more independently, while deeper layers benefit from their integration. We defer further investigation of the gating mechanism such as exploration of input-dependent gates, and vectorized learnable parameters to future work.

**Two-stream hybrid models with reverse gating** We also experiment with dual-stream hybrid models where the Mamba stream edits the hidden states of a Transformer. In this setup, the tanh activation is applied at the outputs of the Mamba model (Figure 2), and we refer to this two-stream model as $\text{Hybrid}_{2SR}$, where the gating function is reversed. While this configuration is unlikely to be implemented in practical scenarios where leveraging the inference benefits of the SSM stream is desirable, it provides valuable insights into the capacity and behavior of hybrid architectures. Following the approach used in our position retrieval experiments, we initialize training with the tanh gating disabled. The results, presented in Figure 19, demonstrate that reversing the gate leads to accelerated convergence, with performance further approximating that of a pure Transformer block. Additionally, $\text{Hybrid}_{2SR}$ exhibits similar per-token performance characteristics throughout training as observed in other hybrid model configurations, suggesting consistent learning dynamics across different gating strategies.

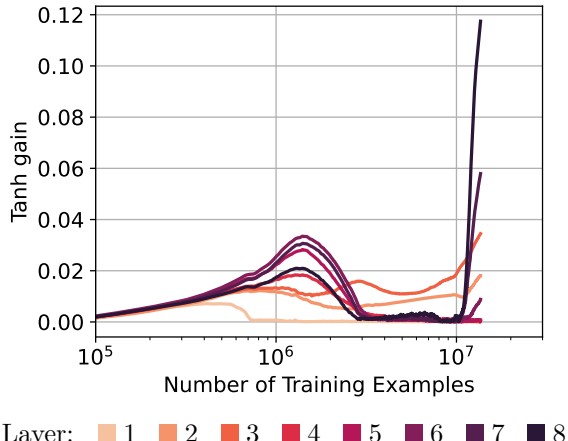

Figure 18: Gating of effect of $\text{Hybrid}_{2S}$ for the task of position retrieval.

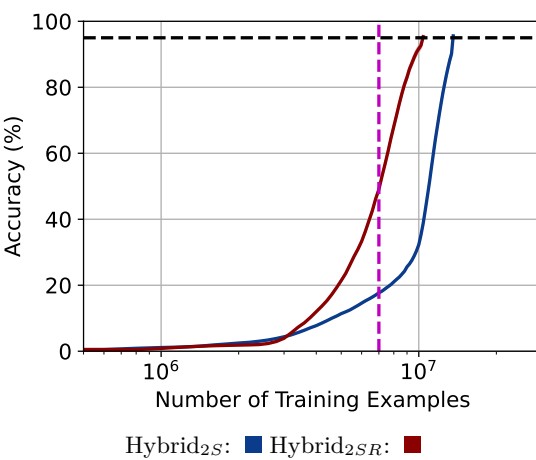 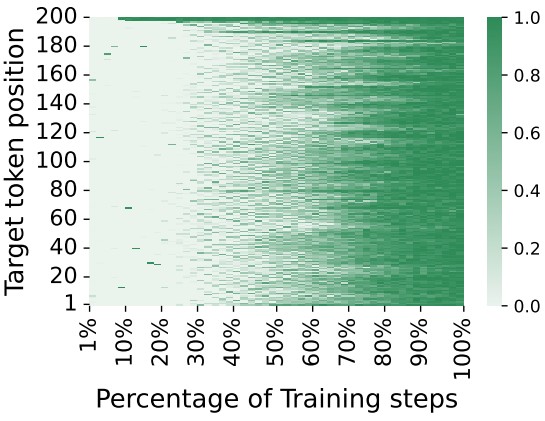

Hybrid$_{2S}$: ■   Hybrid$_{2SR}$: ■

Figure 19: **Left:** Data efficiency on the position retrieval. The two stream hybrid with reverse gating (Hybrid$_{2SR}$), where Mamba output edits the hidden states of a Transformer, converges faster than Hybrid$_{2S}$ and even approximates the performance of the Transformer (RoPE) shown here as a purple dashed line. **Right:** The per-token performance of Hybrid$_{2SR}$.

### B.5 Locality-aware embeddings

Figure 20 shows the embeddings of the positions tokens obtained from all models every 25% of the training projected using PCA. We observe similar findings regarding all models, where they form structured spiral-like representations from early stages of training and deviating from this structure as the training progresses. These results further corroborate the findings presented in Figure 9. Unlike SSMs, hybrid models do not sustain the spiral-like structure across training; however, they still converge to locality-aware representations.

We further demonstrate this property by plotting the cosine similarities of position tokens from all models, including Transformers, and SSMs using $d \in \{32, 64, 128\}$ PCA dimensions, as well as the similarities without any dimensionality reduction. The results are illustrated in Figures 21 and 22. Focusing on the first row of the two figures we observe that all models with SSM blocks learn meaningful, low-dimensionality associations between tokens that describe adjacent positions. Transformers clearly do not have this property as they learn an arbitrary mapping between tokens and positions. Subsequent rows correspond to plots with higher dimension of PCA showing a similar trend. Finally, we can see in the last row of the plots that, without the use of PCA, Transformers assign near-identical values for all positional tokens while SSM blocks maintain locality-aware representations.

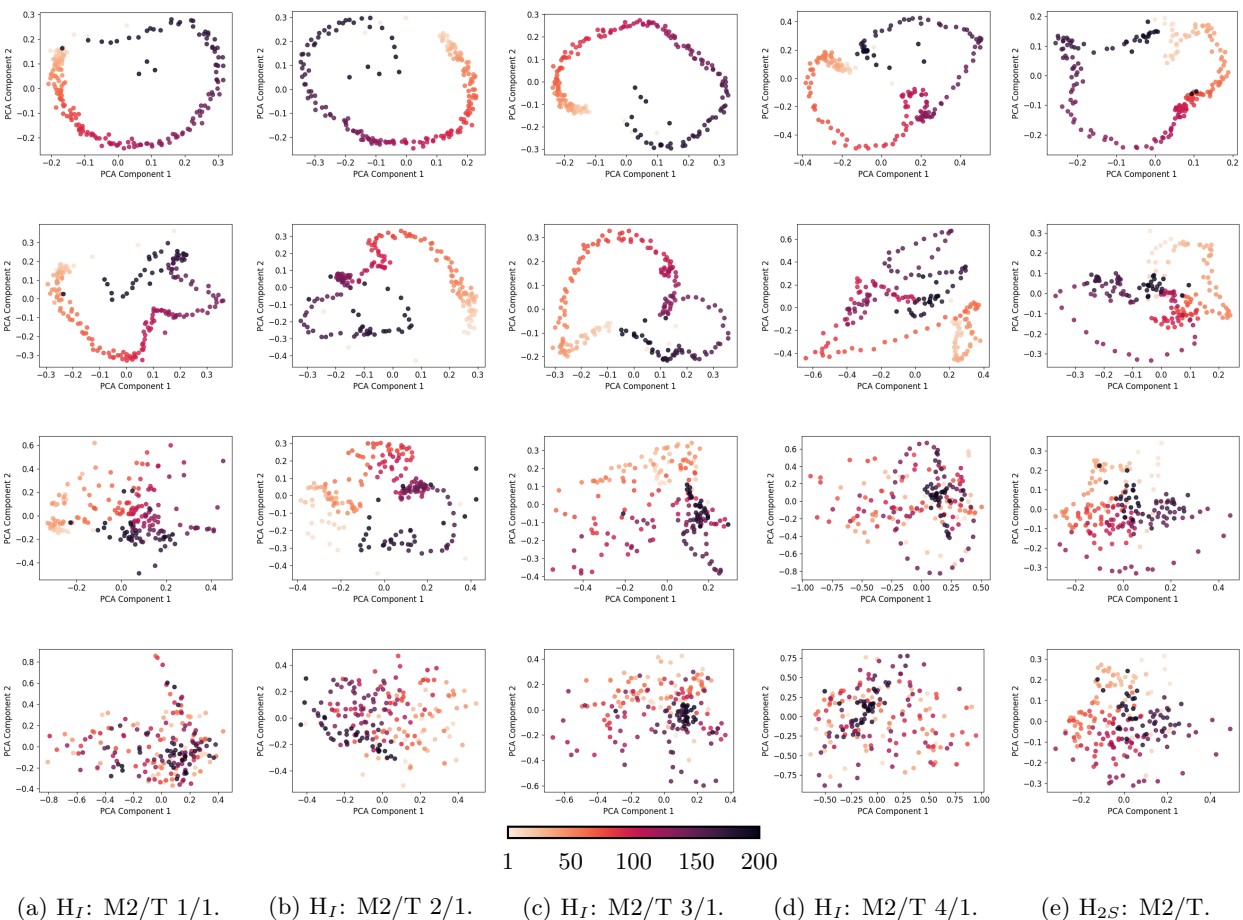

(a) H$_I$: M2/T 1/1.  (b) H$_I$: M2/T 2/1.  (c) H$_I$: M2/T 3/1.  (d) H$_I$: M2/T 4/1.  (e) H$_{2S}$: M2/T.

Figure 20: We track the embeddings of the position tokens in a 2D plane across training for hybrid interleaved (H$_I$) and two-stream (H$_{2S}$) models. Each row corresponds to 25% of the training progress relative to each model, while warmer colors in the spectrum correspond to positional tokens pointing to the start of the sequence. Models with SSM blocks create locality-aware embeddings forming a two-dimensional spiral, while Transformers learn a mapping between the positional token and the actual position in the sequence without any particular structure.

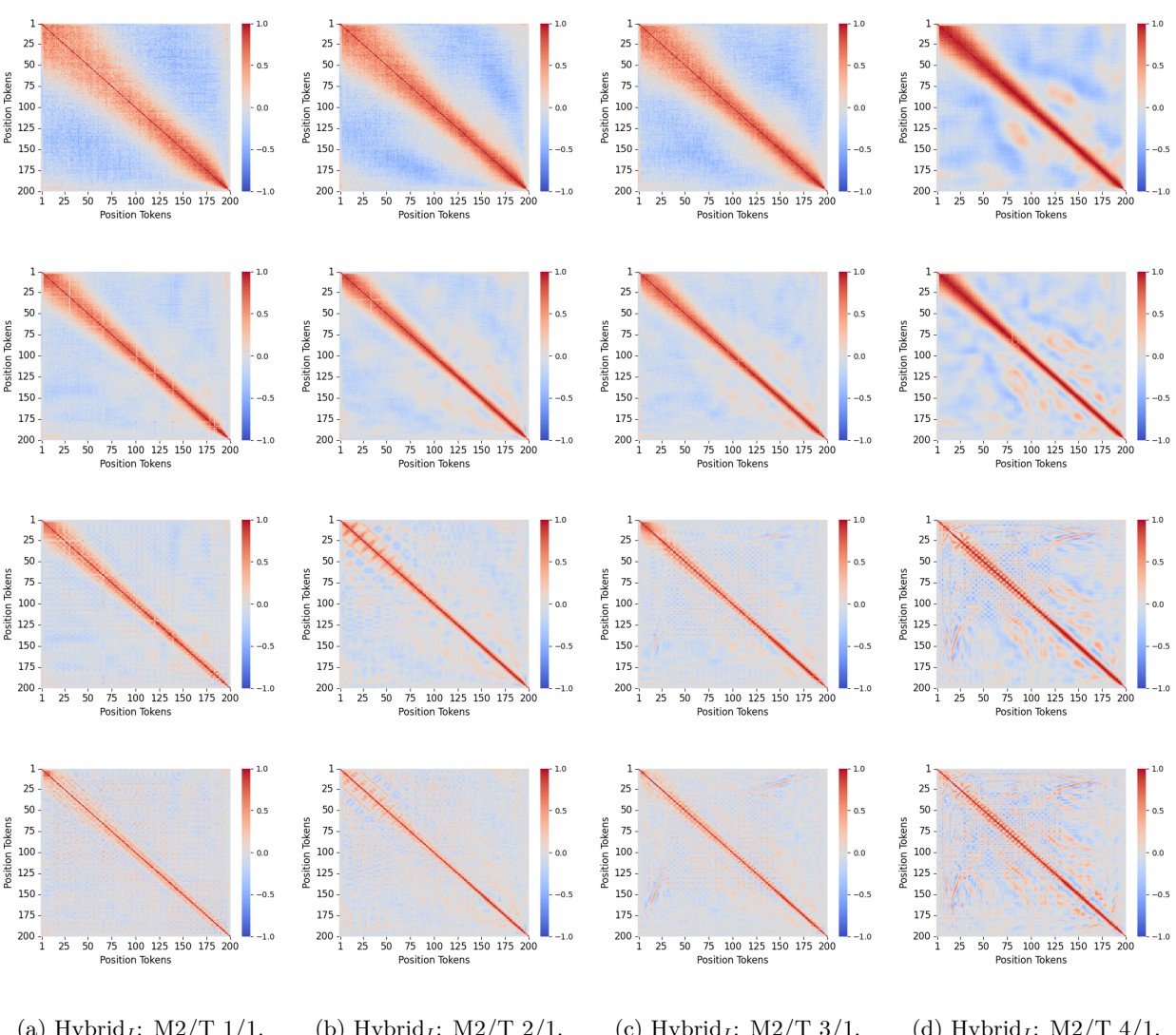

(a) Hybrid$_I$: M2/T 1/1.  (b) Hybrid$_I$: M2/T 2/1.  (c) Hybrid$_I$: M2/T 3/1.  (d) Hybrid$_I$: M2/T 4/1.

Figure 21: We plot the cosine similarities between embeddings of position tokens projected using PCA (dim=32/64/128) in rows one to three, and without any dimensionality reduction shown in the last row. Each column corresponds to a hybrid interleaved model.

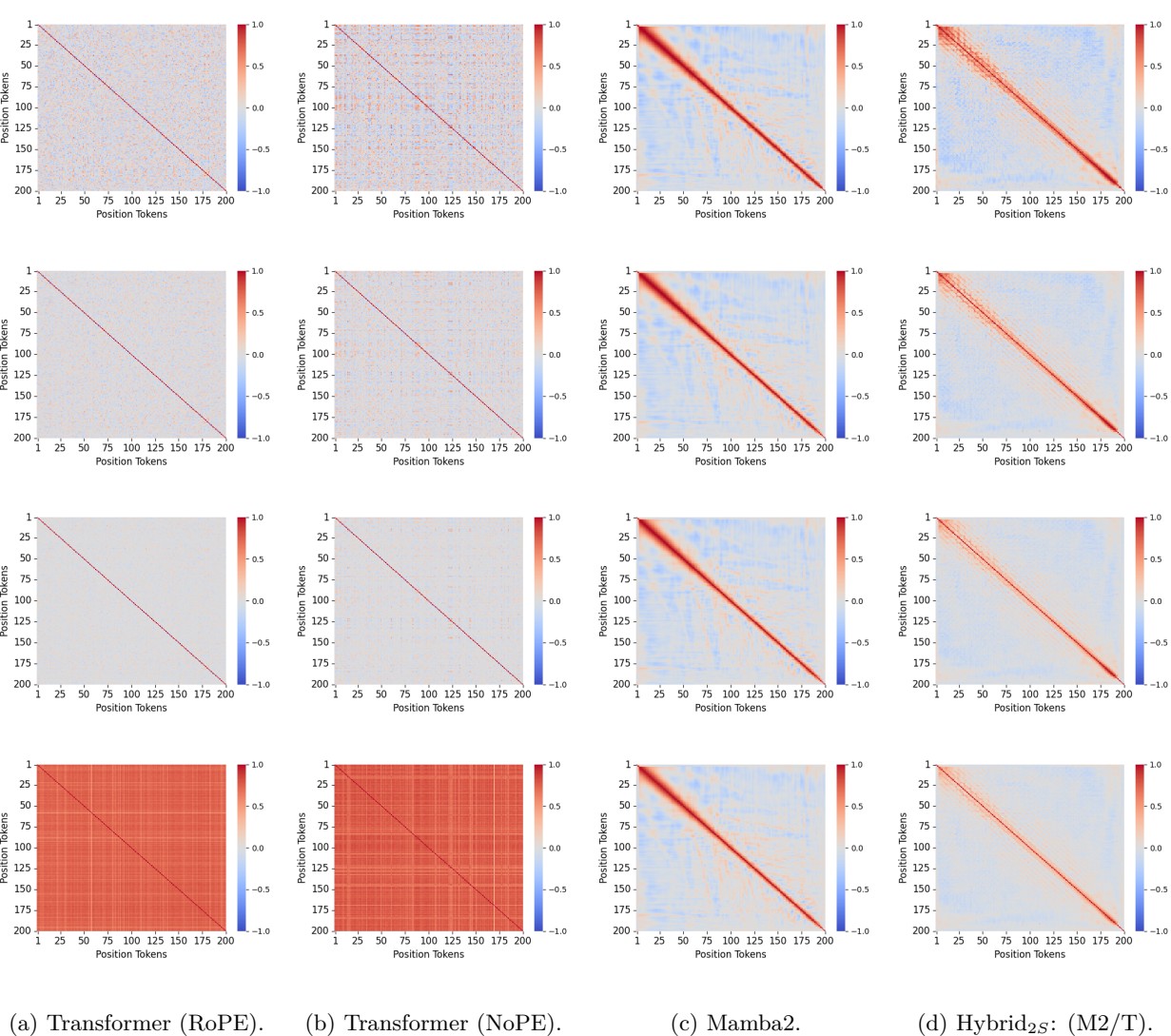

(a) Transformer (RoPE).   (b) Transformer (NoPE).   (c) Mamba2.   (d) Hybrid$_{2S}$: (M2/T).

Figure 22: We plot the cosine similarities between embeddings of position tokens projected using PCA (dim=32/64/128) in rows one to three, and without any dimensionality reduction shown in the last row. **(a)** Transformer with RoPE positional embeddings **(b)** Transformer without positional embeddings, **(c)** Mamba2, and **(d)** Hybrid with interleaved Mamba2 and Transformer blocks.

