# OpenReview forum: "Retrievit: In-context Retrieval Capabilities of Transformers, State Space Models, and Hybrid Architectures"
_TMLR — Accepted by TMLR_

### Review · Reviewer_16tm · 2026-03-14

**Summary Of Contributions:**

The paper compares the performance of Transformers, state-space models and hybrid Transformer-state-space models trained and evaluated on two synthetic tasks:
- n-gram task: Retrieving the k tokens following an a specific sequence of n tokens
- Position task: Retrieving the position of a query token where each position $i$ has a unique output token $p_i$ associated with it

It considers two types of hybrid models -- variants that interleave Transformer and Mamba blocks and variants that run Transformer and Mamba blocks in parallel and sum their outputs mediated through a gate on the Transformer part -- and compares them to pure Transformers and pure Mamba models of roughly equivalent size.
It finds that hybrid models require less training data than pure Transformers or pure Mamba models to solve the n-gram task but more training data than Transformers on the position task.
In addition, it finds that length generalization is better for hybrid models than for pure Transformers or pure Mamba models on the n-gram task.
The paper further analyses the learned output embeddings of the position tokens in the position task and finds that they are spatially structured in Mamba and hybrid models but not in Transformer models.

**Additional Comments:**

Typo on page 2: "Synethetic"

**Audience:**

Yes

**Audience Explanation:**

I find it difficult to assess how interesting the findings in this paper are to others:

- Synthetic context retrieval has been investigated in a number of prior papers for Transformers and SSMs [e.g., 1], what might be novel (but I am not an expert in this literature) is the application to hybrid models
- The position retrieval task that reveals a particular weakness of SSM-based models is interesting (if novel, again I am not fully sure)
- It is unclear what interest in the particular hybrid models investigated here exists given that they are only conceptually linked to existing architectures

[1] Repeat After Me: Transformers are Better than State Space Models at Copying

**Broader Impact Concerns:**

I see no specific concerns for the broader impact of this paper.

**Claims And Evidence:**

No

**Claims Explanation:**

I will evaluate the main claims made in the abstract one by one in the following:

> We find that hybrid models outperform SSMs and match or exceed Transformers in data efficiency and extrapolation for information-dense context retrieval.

I think it needs to be stressed that the experiments are conducted with small models (on the order of 150M parameters) that have been trained from scratch purely on synthetic tasks.
Hybrid models only match or exceed Transformers on the n-gram task but not the position task.
Length generalization has not been tested on the position task.
I think "information-dense context retrieval" should be changed to more clearly reflect that it is referring to the synthetic n-gram task.

> However, Transformers maintain superiority in position retrieval tasks.

The position retrieval task tests whether models can easily retrieve the index of a token in the sequence.
Transformers with positional encodings can do this by reading out the position from the positional encoding matching the query token.
My understanding of Mamba is that it does not use such position encodings and thus this task exposes that reading out the index of a token in the sequence is much more difficult for such models.
The fact that the hybrid models however cannot solve this task as well as the Transformer is puzzling and hints at the fact that the hybrid architectures investigated here cannot efficiently use their Transformer components.
Please see the requested change section for questions/suggestions on this particular issue.

> Through representation analysis, we discover that SSM-based models develop locality-aware embeddings where tokens representing adjacent positions become neighbors in embedding space, forming interpretable structures.

It is true that SSM-based models need to learn some form of spatial encoding if index-based positions need to be retrieved and it is interesting to see this confirmed in the output token embeddings that encode local spatial structure.

> This emergent property, absent in Transformers, explains both the strengths and limitations of SSMs and hybrids for different retrieval tasks.

I think the term 'emergent property' is not fitting in this context.
It is a _learned_ property that directly follows from the task objective.
The fact that it is absent in Transformers is relatively straightforwardly explained given their positional encodings.
(It is less straightforward in the NoPE Transformer setting which serves to confirm the maybe surprising finding that causal Transformers can work without positional encoding.)

I would further say that the notion that this property "explains the strengths and limitations of SSMs for different retrieval tasks." is too imprecise.
This property by itself is not an explanation and I think this argument needs to be made more carefully (see requested changes).

> Our findings provide principled guidance for architecture selection based on task requirements

This claim might be too far fetching giving the limited scope of the paper (small synthetic models).

**Requested Changes:**

### Critical changes to secure my recommendation

> We demonstrate that models with SSM blocks learn a locality-aware mapping, i.e., tokens that depict adjacent positions are neighbors within the embedding space. We show that this is a unique property of models with SSM as Transformers do not converge into such representations.

1. I would suggest to change the framing of this finding. Transformers can directly rely on their positional encoding (RoPE) or directly infer the index in the causal setting (NoPE). The results indicate that SSMs do not have an easy mechanism to do so and need to painstakingly learn position encodings.

> For position retrieval, Transformers maintain the lead while models with SSM blocks lag behind. We attribute this to the locality-aware embeddings that SSM blocks induce, which can be considered as an emergent and interpretable property, but one that ultimately hinders precise two-hop associative lookup.

2. It is unclear to me how the structure of the SSM output embeddings for the different position tokens can explain why they fail? This task reveals that SSMs lack a positional encoding like RoPE or an easy way of inferring position like NoPE. But why can the hybrid models not use their transformer RoPE part to solve this task? Could this be due to a lack of appropriate skip connections? I tried to infer this from the provided source code but found TODO notes in the code (hybrid_par.py, hybrid_seq.py) that suggest the behavior is not fully understood?
```
# src/recallibrate/models/hybrid_par.py
# TODO: Note sure if this is needed
# hidden_states = hidden_states + residual
```

```
src/recallibrate/models/hybrid_seq.py
# TODO: Check if this is true
# I believe that the residual has already been applied in the transformer layer
# So, if the transformer is final block of the model, we don't need to apply the
# residual
# If the mamba is the final block, we need to apply the residual which is done
# internally in the mamba layer
# residual = None
```

> The gate is zero-initialized such that the Transformer stream is inactive at the start of training, allowing the model to progressively learn a balance between global attention and recurrent compression. All hybrid models use RoPE for the Transformer blocks.

3. Why is the gate not chosen to equally weigh the Transformer and SSM block? What effect does that have on the hybrid model performance? What final values do the gates obtain after training, e.g. in Figure 3 when the hybrid-two-stream models outperform non-hybrid models and in Figure 7? Does initializing the gate such that the Transformer input is not attenuated result in the position retrieval task to be solved close to the Transformer performance?

> Following prior work (Jelassiet al., 2024), we define hyperparameters such that all models are matched in parameter size (≈ 150 − 160M
parameters).

4. Are these models approximately compute matched? Please add this information, i.e. how many FLOPs do the different models require, since it helps to contextualize the data efficiency claims.

> Since Transformer blocks have access to all prior tokens, their role is to edit the SSM’s hidden state with information that may have been discarded in a previous timestep.

5. What is the evidence for this statement? Did you conduct experiments to this extent or does prior work make a case for this?


### Non-critical changes, not required to secure my recommendation

- Figure 3 and 7 are difficult to parse: They inconsistently reuse colors between panels, show too many lines per panel and the hybrid types names are not easy to understand. Maybe move "Mamba I" results to the appendix given that no hypothesis pertains to the differences between Mamba I and Mamba II? Maybe make the hybrid models' names more explicit as hybrid-interleaved and hybrid-two-stream?

> The ability to perform a two-hop association by matching the query to its position in the sequence

- I am not sure I would call this a two-hop task. It entierly depends on how location is encoded by the model. Given a positional encoding it is arguably a one-hop task.

> Similarly to test-time extrapolation, we train all models with examples of at most 100 tokens until they reach a perfect score.

- How is it possible to reach a perfect score in the ambiguous setting? How is the score defined? Please clarify this in the manuscript.

- The purple dashed line in Figure 7a is not explained again in this caption but probably indicates the transformer RoPE performance should rather be displayed in 7b, 7c instead?

- The different x-axis scaling/limits in Figure 8 makes it difficult to compare panels

- NoPE-Transformer is missing from Figure 8 but might be insightful to look at: In RoPE the position can be readout directly from the positional encoding, without it, a position encoding must be learned in some way which might explain the sharp transition in performance in the RoPE-transformer across positions.

- Not sure what Figure 10/11 bring beyond Figure 9, maybe move to appendix?

---

> ### Author Response · Authors · 2026-03-18
> **Response to Reviewer 16tm 1/4**
>
> We thank the reviewer for the constructive feedback. We report below our response to the reviewer’s comments:
>
> #### **Re: claims made in the submission supported by accurate, convincing and clear evidence**
>
> 1. **Rephrasing abstract to more accurately reflect the experiments:** We would like to point out that the abstract already states that the models are being trained on purely synthetic tasks. We are happy to further specify this by explicitly stating the size of the models in the study. At the same time, we would like to highlight that this is a feature of work: this specific design choice allows us to make a precise comparison across models of different families trained with different data and training regimes
>
> We also agree that the information-dense context retrieval can be misunderstood. The motivation behind this term was to describe in-context retrieval tasks where the model needs to retrieve precise information from the context (e.g., the n-gram retrieval). We plan to modify our statement as follow:
>
> We find that hybrid models outperform SSMs and match or exceed Transformers in terms of data efficiency and extrapolation for tasks that require precise information retrieval from the input context.
>
> Finally, we will rephrase the last part of the abstract to fit the scope of the paper as:
>
> Notably, causal attention is sufficient for acquiring positional associations, and the introduction of positional encoding amplifies this behavior, leading to a consequent improvement in data efficiency. SSMs on the other hand update their internal representations incrementally and without positional encodings, are required to learn these associations. Our findings reveal fundamental differences in how Transformers and SSMs, and hybrid models learn positional associations.
>
> 2. **Testing length generalization** on the position task is not straightforward as the task by definition requires the model to use index embeddings that exceed the training sequence length (100 tokens). This is because we framed the task in a way that all input tokens appear only once in the sequence to eliminate any ambiguities. An alternative implementation is to allow some of the tokens to appear multiple times and linearly interpolate the learned index embeddings to enable test-time extrapolation. We experimented with this setting and found that no model was able to generalize even at moderate length (110-150 tokens). This finding is perhaps not surprising as naive interpolation without fine-tuning does not enable extrapolation. For this reason, we decided to omit these results from the paper.
>
> 3. **Performance on Transformers/SSMs/Hybrid on Position Retrieval:** We partially agree with the reviewer; Transformers with causal attention can learn the task even without positional information faster than SSMs do. The addition of RoPE embeddings amplifies this behavior. However, the role of RoPE is to influence the dot product between keys and queries. As such Transformers with RoPE do not simply “read” out the position from the token that matches the query but instead try to align the query / key vectors between the query token and its corresponding match in the sequence maximizing their dot-product. Consequently, the value vector of the query token is amplified after softmax scoring.
>
> With regards to the performance of hybrid models, we elaborate on the critical changes section. Overall, the hybrid models have mixed SSM / attention blocks and that synergize well with regards to the n-gram retrieval task as they outperform pure SSM / Transformer models, and so the argument that these models underutilize the attention blocks is not true. As far as the position retrieval task is concerned, we conjecture that it is difficult for models with models with very sparse attention blocks to propagate the effect of RoPE encoding across multiple consecutive residual streams ignoring the SSM pathway. This results in prolonged training compared to hybrid variants with higher Transformer / SSM ratio. This is also verified in Figure 7c where increasing the density of Transformer blocks accelerates the training.

---

> ### Author Response · Authors · 2026-03-18
> **Response to Reviewer 16tm 2/4**
>
> #### **Re: Would at least some individuals in TMLR's audience be interested in knowing the findings of this paper?**
>
> We have already acknowledged that these tasks have been established by prior works (Section 2 and Section 3.1). We are extending this line of work by measuring the capabilities of hybrid models relative to pure Transformers and SSMs, which is absent from prior work. Regarding the particular selection of hybrid models, we have already explained this in Section 3.2. In particular, the design is inspired by larger models [1,2] following an interleaved or a two-stream setup [3]. These models have shown promising sequence modeling capabilities but comparing them on equal grounds is not straightforward as they use different training mixtures, regimes, and may use different model weights as starting checkpoints. Furthermore, their hybrid configurations are not fully intuitive. For instance [1] briefly elaborates on the ratio of Attention-to-Mamba layers while [2] imitates configurations from prior work by only looking at validation loss curves and ignoring actual recall capabilities.
>
> [1]: Lieber, Opher, et al. "Jamba: A hybrid transformer-mamba language model." arXiv preprint arXiv:2403.19887 (2024).
>
> [2]: Blakeman, Aaron, et al. "Nemotron-h: A family of accurate and efficient hybrid mamba-transformer models." arXiv preprint arXiv:2504.03624 (2025).
>
> [3]: Dong, Xin, et al. "Hymba: A hybrid-head architecture for small language models." arXiv preprint arXiv:2411.13676 (2024).

---

> ### Author Response · Authors · 2026-03-18
> **Response to Reviewer 16tm 3/4**
>
> #### **Re: Critical changes to secure my recommendation**
>
> 1. **Reframing the finding:** As already explained in the first section, we will clarify our paper’s position and rephrase our claims to better reflect the findings of our study.
>
> 2. **Why can the hybrid models not use their transformer RoPE part to solve this task?** Let us first apologize for the submitted supplementary material. This was a mistake when clearing up the code and removing old/redundant pieces. The issue is that gpt-neox/mamba libraries have different orders of residual and layer norm in each block.    Instead of LN -> Attn / MLP -> Add, Mamba does Add -> LN -> Mamba Block returning both the residual and the main branch. We have updated the supplementary material and hope this clarifies the confusion.
>
> To answer the question, we note that the RoPE embeddings operate on the queries and keys before the softmax activation. Furthermore, the rotation angle is relative to the distance between two tokens and thus the positional information does not explicitly encode absolute positions. Taken together, the RoPE embeddings only affect the softmax scoring without impacting the values and helps the model converge faster compared to the NoPE variant.
>
> In the context of hybrid models, this information could survive through the residual stream. In practice, though, due to sparse attention blocks, it might be hard to propagate this information across multiple residual streams, especially when the main branch is composed of SSM blocks without positional encodings. As already mentioned (point 3 in the first section), increasing the density of Transformers within the hybrid interleaved variant results in accelerated training, which verifies the above assumption.
>
> 3. **Experiments with different gate variants:** The design idea behind the gate stems from the fact that if the Mamba stream is sufficient, we would prefer to avoid quadratic cost of the attention stream during inference. As such, we focused primarily on gates where the Transformer stream is inactive during initialization. However, we also explored the reverse direction, where the Transformer stream is active. The results are presented in section B.4 where we observe that the reversed model behaves closer to a Transformer (RoPE) model.
>
> 4. **FLOPs of each model:** We would like to clarify that we use a definition of data efficiency that is not related to data efficiency as intended in LLM Scaling Laws [1]. In this paper, we are interested in a definition of data efficiency that focuses on understanding how much does a model (where every model studied has roughly the same size N) learn per example seen during the training given a fixed training dataset rather than measuring the compute-optimal training setup for a given model family. This is the reason why our plots show the accuracy gain over the course of training examples. Regardless, we approximate the FLOPs required for a single forward pass for Transformers, Mamba, and Hybrid models using an empirical estimate from [1], and the FLOPs formula for mamba [2]. All models have require approximately 30 x 10^9 FLOPs for a single forward pass assuming an input sequence of 100 tokens
>
> 5. **Evidence on the role of Transformer in correcting the hidden state** By design, at each timestep Mamba may choose to update its hidden state with information from the current token. This update is incremental; it only depends on the previous hidden state and the input value at the current timestep. Transformers, on the other hand, can perform updates with access to all prior tokens via self-attention. Therefore, a Transformer layer within the hybrid model can edit the hidden state at the present timestep with information from a previous hidden state that was potentially discarded.
>
> While we have not conducted any experiments to show this (we are also curious how one can even formulate and study this problem), prior work also points to similar behavior (Please see Section 6.2 in [3]). In this case , the performance of Mamba is tied to the ability to perform the induction heads task, which is closely aligned to the tasks that we also experimented with.
>
> [1]: Kaplan, Jared, et al. "Scaling laws for neural language models." arXiv preprint arXiv:2001.08361 (2020).
>
> [2]: https://github.com/state-spaces/mamba/issues/110#issuecomment-1919470069
>
> [3]: Lieber, Opher, et al. "Jamba: A hybrid transformer-mamba language model." arXiv preprint arXiv:2403.19887 (2024).

---

> > ### Author Response · Authors · 2026-03-18
> > **Reviewer 16tm 4/4**
> >
> > #### **Re: Non-critical changes, not required to secure my recommendation**
> >
> > 1. **Readability of figures:** We will update figures 3 and 7 to be easier to read. We are happy to move the Mamba results to the appendix, given that our focus is primarily on Mamba 2 models. The purple lines are going to be corrected in the revised version.
> >
> > 2. **Perfect score in duplicate queries** We clarify that the duplicate queries are only present during evaluation to facilitate ambiguity. During training / validation all examples have a single correct response.
> >
> > To avoid any confusion with the versions of the submitted pdf we would appreciate it if we could update it once we have received all 3 reviews.

---

> > ### Comment · Reviewer_16tm · 2026-04-12
> > **Follow-up questions**
> >
> > Thank you for your detailed response. Please allow me two follow-up questions
> >
> > > Experiments with different gate variants: The design idea behind the gate stems from the fact that if the Mamba stream is sufficient, we would prefer to avoid quadratic cost of the attention stream during inference. As such, we focused primarily on gates where the Transformer stream is inactive during initialization. However, we also explored the reverse direction, where the Transformer stream is active. The results are presented in section B.4 where we observe that the reversed model behaves closer to a Transformer (RoPE) model.
> >
> > It is unclear to me how the quadratic cost of the attention stream would be avoided during inference as long as the multi-head attention layer is present with a non-zero gate in the hybrid model?
> >
> > > Evidence on the role of Transformer in correcting the hidden state By design, at each timestep Mamba may choose to update its hidden state with information from the current token. This update is incremental; it only depends on the previous hidden state and the input value at the current timestep. Transformers, on the other hand, can perform updates with access to all prior tokens via self-attention. Therefore, a Transformer layer within the hybrid model can edit the hidden state at the present timestep with information from a previous hidden state that was potentially discarded.
> >
> > I would suggest to soften this claim as long as there is no empirical evidence for it. There are many other explanations in theory of how these two types of blocks could work together, for instance Mamba blocks could just function as a no-op and let the Transformer blocks do all the work.

---

> > > ### Author Response · Authors · 2026-04-13
> > > **Response to follow-up questions**
> > >
> > > We appreciate that the reviewer engaged in discussion.
> > >
> > > **1. Clarification on the two-stream response.**
> > > In principle, because the Transformer stream is inactive during initialization, then it is possible that at some layers the entire sequence is processed exclusively through the Mamba block. As such after training one could completely ignore the Transformer blocks at these types of layers. In our case we observed a strong gating effect in deeper layers and a weaker one in early layers of the model.
> > >
> > > **2. Role of Transformer in a Hybrid Block**
> > > We rephrased this part to better reflect that this is one possible explanation.

---

### Review · Reviewer_zoFH · 2026-03-17

**Summary Of Contributions:**

This paper presents a systematic analysis of in-context retrieval performance of Transformers, SSMs, and hybrid models. The paper designed experiments for training transformers, SSMs, and hybrid models in fixed settings for N-gram retrieval, and positional retrieval tasks and conducted empirical studies on metrics including data efficiency, length generalization, and duplicate queries robustness and learning dynamics. Experiments results show different performance and limitations of these models on N-gram retrieval, and positional retrieval tasks, suggesting that hybrid models could be a promising alternative to transformers.

**Audience:**

Yes

**Audience Explanation:**

SSMs and hybrid models have been a popular research area as an alternative to transformer models. This paper provides detailed analysis that help understand the performance and limitations of these models.

**Broader Impact Concerns:**

No concerns.

**Claims And Evidence:**

Yes

**Claims Explanation:**

The paper presents finding 1-6, evidenced by detailed experiment evaluation and concluded with discussions about advantage and limitations of these models on n-gram and positional retrieval.

**Requested Changes:**

The experiments are in small scale (e.g 100 or 200 tokens sequence length in n-gram and positional retrieval). Need to clarify whether this setup is sufficient.

---

> ### Author Response · Authors · 2026-03-23
> **Response to Reviewer zoFH**
>
> We thank the reviewer for their feedback. We appreciate that they found our experiments thorough concerning the benefits of these models across different in-context retrieval tasks.
>
> We would like to point out that we use similar sequences as prior [1, 2] across all of our experiments. We experimented with larger sequences but in many cases the models were not converging. We hypothesize that this is due to the fact that the sequences are randomly sampled from a fixed vocabulary, i.e the tokens do not contain any meaningful language priors like in standard language modeling. Regardless, the performance on these types of tasks has been shown to be indicative of the sequence modeling capabilities of larger models. We will update section 3.1 concerning the task overview to more accurately reflect this.
>
> [1] Jelassi, S., Brandfonbrener, D., Kakade, S.M. and Malach, E., 2024. Repeat after me: Transformers are better than state space models at copying. arXiv preprint arXiv:2402.01032.
>
> [2] Pantazopoulos, G., Nikandrou, M., Suglia, A., Lemon, O. and Eshghi, A., 2024. Shaking Up VLMs: Comparing Transformers and Structured State Space Models for Vision & Language Modeling. arXiv preprint arXiv:2409.05395.

---

### Review · Reviewer_D94e · 2026-04-06

**Summary Of Contributions:**

The paper studies Transformers (with RoPE and NoPE), State Space Models (Mamba, Mamba2), and hybrid architectures (Interleaved and two-stream model) that combine the two, on synthetic in-context retrieval tasks to determine how the architecture impacts in-context retrieval capabilities. They train and evaluate the models on two tasks: n-gram retrieval, where the model must copy tokens following a query pattern, and position retrieval, where it must look up a token's positional index in the sequence. They assess each architecture under controlled conditions, examining factors like data efficiency, generalization, and learned representations. Furthermore, they conduct representation analysis to understand how each architecture internally encodes positional information. They find that the choice of architecture significantly impacts retrieval behavior: hybrids excel at n-gram retrieval, converging robustly and generalizing to longer sequences, while Transformers lead on position retrieval. They trace this divergence to fundamentally different learned representations, SSMs develop structured, locality-preserving embeddings while Transformers learn unconstrained mappings.

**Audience:**

Yes

**Audience Explanation:**

The paper addresses questions that are directly relevant to researchers working on in-context learning and sequence models.

**Claims And Evidence:**

Yes

**Claims Explanation:**

The core comparative claims (which architecture is better on which task, data efficiency rankings, length generalization results) are well-supported by exhaustive experiments and clear experimental evidence under controlled conditions (All models are roughly matched in parameter count, trained and evaluated under identical conditions with multiple seeds and learning rate sweeps on synthetic tasks). In addition, the representation analysis is descriptive and visually compelling. The authors are transparent about their contributions, acknowledging that their findings are grounded in synthetic settings and that validation on real-world tasks at larger scales remains important future work.

**Requested Changes:**

I would like to thank the authors for their thorough and well-structured investigation. Below are my requested changes and clarifications:

- I find Figure 1 to be a bit unclear. As per my understanding, in the suffix setting the query appears at the end of the input, and the model must output 'k' tokens. So I am unsure why the k tokens are part of the input sequence in the diagram. Similarly, for the position retrieval task, it is a bit confusing to see the position token as part of the input. Maybe a note in the caption distinguishing the training view from the inference-time input/output split would have helped avoid confusion.
- "only penalize the model for mistakes on the expected response and not on the inputs" - Is this representative of language model pretraining in practice, as models are usually trained across the entire sequence? I imagine the gradients would be more noisy if you did train across the sequence, would the results still remain consistent if trained across the sequence?
- Error rates are much higher for prefix compared to suffix in duplicate queries experiment, do the authors have any ideas behind why this might be the case?
- How are the position tokens converted into embeddings from the models?
- In Figure 4, only Mamba2 is included as the SSM baseline, while Mamba is absent despite being evaluated in other experiments. Additionally, although the legend in Figure 4a lists Hybrid2S (Mamba2) as one of the models, its corresponding curve is not clearly identifiable in Figure 4c. Finally, the x-axis in Figures 4b and 4c is labeled 'Number of Input Characters' rather than tokens, can the authors clarify if each character is a token.
- Are the decoding strategy and hyperparameters the same for all experiments (greedy and temperature 0)?
- Can the authors provide more information on the synthetic data used for both tasks (vocabulary size and token types)?
- The authors show that hybrid models achieve "near-perfect length generalization", it would be worth asking at how long a sequence length this advantage fades.
- The authors mention running multiple seeds, but most figures lack error bars or confidence intervals, do the reported results reflect the best run, an average, or some other aggregation?

---

> ### Author Response · Authors · 2026-04-09
> **Response to Reviewer D94e**
>
> We appreciate that the reviewer found our investigation thorough and well structured. Below we elaborate on the requested changes:
>
> **1. Confusion about inputs/outputs figure 1.**
> This figure represents the training view where the outputs are shifted to the left, we have now updated the caption to reflect this.
>
> **2. Masking input and only penalizing expected response**
> One could view this as a standard SFT stage during the development of a language model where the inputs correspond to a prompt and the outputs (the k-tokens for our synthetic tasks) correspond to the expected response of the language model.
>
> We did not train across the entire sequence mostly because, as the reviewer also suggests, the gradients would become very noisy. However, we estimate a similar behavior, perhaps with an offset of some training updates to account for the disentanglement of the noisy gradients and the actual model output.
>
> **3. Error rates are much higher for prefix compared to suffix in duplicate queries experiment**
> That is a very interesting question, we were also puzzled about this behavior but we hypothesize the following:
>
> In the prefix version, by the time the model needs to retrieve the n-gram following the query, it has already processed multiple prior instances of the same query in its attention in the case of Transformers, or in the hidden state in the case of SSMs.  For Transformers, the softmax over prior tokens creates competition between these positions at the time of retrieval (last input token) but also when updating the representations for the input sequence (as duplicate instances can attend to the prefix). Consequently, the model cannot cleanly identify which duplicate instance to anchor to for retrieval.  In the suffix case the softmax introduces competition between the duplicate queries only at the time of retrieval, which is much closer to the training condition and so the error rate stays low. A similar behavior could also explain the high error rates in the case of SSMs, where the duplicates compete for the hidden state, though as Fig 6 shows, Mamba models tend to heavily prioritize the last duplicate occurrence of the query.
>
> We have now added this as a justification in the appendix rather than in the main paper to avoid any bold claims.
>
> **4. How are the position tokens converted into embeddings from the models?**
> These are regular tokens that are mapped to embeddings via a standard embedding matrix similar to all other tokens in the sequence
>
> **5. Decoding strategies**
> All experiments use identical conditions (greedy decoding + 0 temperature)
>
> **6. more information on the synthetic data used for both tasks**
> We added a small section in the appendix explaining these details and re-iterating any information that was introduced in the main paper. In particular:
>
> **N-gram retrieval** All experiments are conducted using a vocabulary of 30, the size of the query n-gram is $n=2$ and the output k-gram is $k=3$.
> Additionally, all sequences during training and evaluation are randomly sampled from the vocabulary with replacement ensuring that the query n-gram appears only once within the sequence.
> Note that this does not prevent the existence of other unique n-grams.
> With regards to the duplicate experiments, we ensured that the query appears multiple times and each duplicate is followed by a unique k-gram.
>
> **Position retrieval** All experiments are conducted using a vocabulary of 200 tokens describing the input sequence and an additional 200 tokens for their position. Each sequence during training and evaluation is constructed by randomly shuffling the input 200 tokens.
>
> **7. hybrid models achieve "near-perfect length generalization**
> We have updated the generalization experiments testing up to 1k sequence length where we observe that the performance of hybrid models start to deteriorate.
>
> **8. figures lack error bars or confidence intervals**
> For the efficiency / extrapolation curves we selected the top 3 runs for each model and plot the average performance with min/max margins. For the curves with the per-token performance and the visualization of embeddings we used the best run (in terms of convergence speed) for each model.

---

### Author Response · Authors · 2026-04-15

Dear reviewers,

We really appreciate that you took some time to review our paper. We have replied to each of your requests and we have also updated the version of the pdf accordingly. Since the discussion period is soon coming to a close (20 April) we are wondering if your concerns are addressed or if there is something else that needs to be clarified.

---

### Decision · Action_Editor_B9vz · 2026-05-26

**Recommendation:** Accept as is

**Audience:**

Yes

**Audience Explanation:**

Hybrid architectures combining softmax self-attention with linear recurrent/SSM-type layers are increasingly adopted for sequence modeling but they are still not very well understood. This paper analyzes and compares such hybrids against their pure counterparts in a toy setup aimed at understanding in-context retrieval. This setup is well designed, in particular because the pure transformer is believed to set the gold standard for retrieval tasks. The fact that hybrids can in some cases outperform transformers in such tasks is an interesting finding, worth sharing with the TMLR readership.

**Claims And Evidence:**

Yes

**Claims Explanation:**

The reviewers found the paper to be generally well presented, and did not find any critical execution flaws.
The authors are encouraged to take the remaining suggestions of the reviewers seriously and incorporate them into the final version of their paper.